# GUARD: A Safe Reinforcement Learning Benchmark

**Weiye Zhao**$^*$                                                          *weiyezha@andrew.cmu.edu*
*Robotics Institute*
*Carnegie Mellon University*

**Yifan Sun**$^*$                                                          *yifansu2@andrew.cmu.edu*
*Robotics Institute*
*Carnegie Mellon University*

**Feihan Li**                                                          *feihanl@andrew.cmu.edu*
*Robotics Institute*
*Carnegie Mellon University*

**Rui Chen**                                                          *ruic3@andrew.cmu.edu*
*Robotics Institute*
*Carnegie Mellon University*

**Ruixuan Liu**                                                          *ruixuanl@andrew.cmu.edu*
*Robotics Institute*
*Carnegie Mellon University*

**Tianhao Wei**                                                          *twei2@andrew.cmu.edu*
*Robotics Institute*
*Carnegie Mellon University*

**Changliu Liu**                                                          *cliu6@andrew.cmu.edu*
*Robotics Institute*
*Carnegie Mellon University*

**Reviewed on OpenReview:** *https://openreview.net/forum?id=kZFKwApeQO*

## Abstract

Due to the trial-and-error nature, it is typically challenging to apply RL algorithms to safety-critical real-world applications, such as autonomous driving, human-robot interaction, robot manipulation, etc, where such errors are not tolerable. Recently, safe RL (*i.e.*, constrained RL) has emerged rapidly in the literature, in which the agents explore the environment while satisfying constraints. Due to the diversity of algorithms and tasks, it remains difficult to compare existing safe RL algorithms. To fill that gap, we introduce GUARD, a **G**eneralized **U**nified S**A**fe **R**einforcement Learning **D**evelopment Benchmark. GUARD has several advantages compared to existing benchmarks. First, GUARD is a generalized benchmark with a wide variety of RL agents, tasks, and safety constraint specifications. Second, GUARD comprehensively covers state-of-the-art safe RL algorithms with self-contained implementations. Third, GUARD is highly customizable in tasks and algorithms. We present a comparison of state-of-the-art on-policy safe RL algorithms in various task settings using GUARD and establish baselines that future work can build on.

---

$^*$Equal Contribution

# 1 Introduction

Reinforcement learning (RL) has achieved tremendous success in many fields over the past decades. In RL tasks, the agent explores and interacts with the environment by trial and error, and improves its performance by maximizing the long-term reward signal. RL algorithms enable the development of intelligent agents that can achieve human-competitive performance in a wide variety of tasks, such as games (Mnih et al., 2013; Silver et al., 2018; OpenAI et al., 2019; Vinyals et al., 2019), manipulation (Popov et al., 2017; Chen et al., 2023; Agostinelli et al., 2019; Shek et al., 2022), autonomous driving (Isele et al., 2019; Kiran et al., 2022; Gu et al., 2022a), robotics (Kober et al., 2013; Brunke et al., 2022), and more. Despite their outstanding performance in maximizing rewards, recent works (García & Fernández, 2015; Gu et al., 2022b; Zhao et al., 2023) focus on the safety aspect of training and deploying RL algorithms due to the safety concern in real-world safety-critical applications, *e.g.*, human-robot interaction, autonomous driving, etc. As safe RL topics emerge in the literature, it is crucial to employ a standardized benchmark for comparing and evaluating the performance of various safe RL algorithms across different applications, ensuring a reliable transition from theory to practice. A benchmark includes 1) algorithms for comparison; 2) environments to evaluate algorithms; 3) a set of evaluation metrics, etc. There are benchmarks for unconfined RL and some safe RL, but not comprehensive enough (Duan et al., 2016; Brockman et al., 2016; Ellenberger, 2018–2019; Yu et al., 2019; Osband et al., 2020; Tunyasuvunakool et al., 2020; Dulac-Arnold et al., 2020; Zhang et al., 2022a).

To create a robust safe RL benchmark, we identify three essential pillars. Firstly, the benchmark must be **generalized**, accommodating diverse agents, tasks, and safety constraints. Real-world applications involve various agent types (e.g., drones, robot arms) with distinct complexities, such as different control degrees-of-freedom (DOF) and interaction modes (e.g., 2D planar or 3D spatial motion). The performance of algorithms is influenced by several factors, including variations in robots (such as observation and action space dimensions), tasks (interactive or non-interactive, 2D or 3D), and safety constraints (number, trespassibility, movability, and motion space). Therefore, providing a comprehensive environment to test the generalizability of safe RL algorithms is crucial.

Secondly, the benchmark should be **unified**, overcoming discrepancies in experiment setups prevalent in the emerging safe RL literature. A unified platform ensures consistent evaluation of different algorithms in controlled environments, promoting reliable performance comparison. Lastly, the benchmark must be **extensible**, allowing researchers to integrate new algorithms and extend setups to address evolving challenges. Given the ongoing progress in safe RL, the benchmark should incorporate major existing works and adapt to advancements. By encompassing these pillars, the benchmark provides a solid foundation for addressing these open problems in safe RL research.

In light of the above-mentioned pillars, this paper introduces GUARD, a **G**eneralized **U**nified S**A**fe **R**einforcement Learning **D**evelopment Benchmark. In particular, GUARD is developed based upon the Safety Gym (Ray et al., 2019b), SafeRL-Kit (Zhang et al., 2022a) and SpinningUp (Achiam, 2018). Unlike existing benchmarks, GUARD pushes the boundary beyond the limit by significantly extending the algorithms in comparison , types of agents and tasks, and safety constraint specifications. The code is available on Github[1]. The contributions of this paper are as follows:

1. **Generalized benchmark with a wide range of agents.** GUARD genuinely supports **11** different agents, covering the majority of real robot types.

2. **Generalized benchmark with a wide range of locomotion tasks.** GUARD comprehensively supports **7** distinct robot locomotion task specifications, which can be combined to represent a wide spectrum of real-world robot tasks that necessitate intricate locomotion for successful completion.

3. **Generalized benchmark with a wide range of safety constraints.** GUARD genuinely supports **8** distinct safety constraint specifications. These included constraint options comprehensively cover the safety requirements encountered in real-world applications.

4. **Unified benchmarking platform with comprehensive coverage of safe RL algorithms.** Guard implements **8** state-of-the-art on-policy safe RL algorithms following a unified code structure.

---

[1]https://github.com/intelligent-control-lab/guard

5. **Highly customizable benchmarking platform.** GUARD features a modularized design that enables effortless customization of new robot locomotion testing suites with self-customizable agents, tasks, and constraints. The algorithms in GUARD are self-contained, with a consistent structure and independent implementations, ensuring clean code organization and eliminating dependencies between different algorithms. This self-contained structure greatly facilitates the seamless integration of new algorithms for further extensions.

## 2 Related Work

**Open-source Libraries for Reinforcement Learning Algorithms**  Open-source RL libraries are code bases that implement representative RL algorithms for efficient deployment and comparison. They often serve as backbones for developing new RL algorithms, greatly facilitating RL research. We divide existing libraries into two categories: (a) safety-oriented RL libraries that support safe RL algorithms, and (b) general RL libraries that do not. Among safety-oriented libraries, Safety Gym (Ray et al., 2019b) is the most famous one with highly configurable tasks and constraints but only supports three safe RL methods. SafeRL-Kit (Zhang et al., 2022a) supports five safe RL methods while missing some key methods such as CPO (Achiam et al., 2017). Bullet-Safety-Gym (Gronauer, 2022) offers support for CPO but is limited in its overall safe RL support, encompassing a total of four methods. Compared to the above libraries, our proposed GUARD doubles the support to eight methods in total, covering a wider spectrum of general safe RL research. General RL libraries, on the other hand, can be summarized according to their backend into PyTorch (Achiam, 2018; Weng et al., 2022; Raffin et al., 2021; Liang et al., 2018), Tensorflow (Dhariwal et al., 2017; Hill et al., 2018), Jax (Castro et al., 2018; Hoffman et al., 2020), and Keras (Plappert, 2016). In particular, SpinningUp (Achiam, 2018) serves as the major backbone of our GUARD benchmark on the safety-agnostic RL portion.

**Benchmark Platform for Safe RL Algorithms**  To facilitate safe RL research, the benchmark platform should support a wide range of task objectives, constraints, and agent types. Among existing work, the most representative one is Safety Gym (Ray et al., 2019b) which is highly configurable. However, Safety Gym is limited in agent types in that it does not support high-dimensional agents (e.g., drone and arm) and lacks tasks with complex interactions (e.g., chase and defense). Moreover, Safety Gym only supports naive contact dynamics (e.g., touch and snap) instead of more realistic cases (e.g., objects bouncing off upon contact) in contact-rich tasks. Safe Control Gym (Yuan et al., 2022) is another open-source platform that supports very simple dynamics (i.e., cartpole, 1D/2D quadrotors) and only supports navigation tasks. Bullet Safety Gym (Gronauer, 2022) provides high-fidelity agents, but the types of agents are limited, and they only consider navigation tasks. Safety-Gymnasium (Ji et al., 2023) provides a rich array of safety RL task categories. However, it faces limitations in implementing / supporting modern safe RL algorithms and offering a flexible testing suite for each category. These weaknesses are particularly notable in the context of locomotion tasks, where there are very limited options for robots, constraints, and objectives. Compared to the above platforms, our GUARD supports a much wider range of robot locomotion task objectives (e.g., 3D reaching, chase and defense) with a much larger variety of eight agents including high-dimensional ones such as drones, arms, ants, and walkers.

## 3 Preliminaries

**Markov Decision Process**  An Markov Decision Process (MDP) is specified by a tuple $(\mathcal{S}, \mathcal{A}, \gamma, \mathcal{R}, P, \rho)$, where $\mathcal{S}$ is the state space, and $\mathcal{A}$ is the control space, $\mathcal{R} : \mathcal{S} \times \mathcal{A} \to \mathbb{R}$ is the reward function, $0 \leq \gamma < 1$ is the discount factor, $\rho : \mathcal{S} \to [0, 1]$ is the starting state distribution, and $P : \mathcal{S} \times \mathcal{A} \times \mathcal{S} \to [0, 1]$ is the transition probability function (where $P(s'|s, a)$ is the probability of transitioning to state $s'$ given that the previous state was $s$ and the agent took action $a$ at state $s$). A stationary policy $\pi : \mathcal{S} \to \mathcal{P}(\mathcal{A})$ is a map from states to a probability distribution over actions, with $\pi(a|s)$ denoting the probability of selecting action $a$ in state $s$. We denote the set of all stationary policies by $\Pi$. Suppose the policy is parameterized by $\theta$; policy search algorithms search for the optimal policy within a set $\Pi_\theta \subset \Pi$ of parameterized policies.

The solution of the MDP is a policy $\pi$ that maximizes the performance measure $\mathcal{J}(\pi)$ computed via the discounted sum of reward:

$$\mathcal{J}(\pi) = \mathbb{E}_{\tau \sim \pi}\left[\sum_{t=0}^{\infty} \gamma^t \mathcal{R}(s_t, a_t, s_{t+1})\right], \tag{1}$$

where $\tau = [s_0, a_0, s_1, \cdots]$ is the state and control trajectory, and $\tau \sim \pi$ is shorthand for that the distribution over trajectories depends on $\pi$: $s_0 \sim \mu, a_t \sim \pi(\cdot|s_t), s_{t+1} \sim P(\cdot|s_t, a_t)$. Let $R(\tau) \doteq \sum_{t=0}^{\infty} \gamma^t \mathcal{R}(s_t, a_t, s_{t+1})$ be the discounted return of a trajectory. We define the on-policy value function as $V^\pi(s) \doteq \mathbb{E}_{\tau \sim \pi}[R(\tau)|s_0 = s]$, the on-policy action-value function as $Q^\pi(s, a) \doteq \mathbb{E}_{\tau \sim \pi}[R(\tau)|s_0 = s, a_0 = a]$, and the advantage function as $A^\pi(s, a) \doteq Q^\pi(s, a) - V^\pi(s)$.

**Constrained Markov Decision Process**  A constrained Markov Decision Process (CMDP) is an MDP augmented with constraints that restrict the set of allowable policies. Specifically, CMDP introduces a set of cost functions, $C_1, C_2, \cdots, C_m$, where $C_i : \mathcal{S} \times \mathcal{A} \times \mathcal{S} \to \mathbb{R}$ maps the state action transition tuple into a cost value. Similar to equation 1, we denote $\mathcal{J}_{C_i}(\pi) = \mathbb{E}_{\tau \sim \pi}[\sum_{t=0}^{\infty} \gamma^t C_i(s_t, a_t, s_{t+1})]$ as the cost measure for policy $\pi$ with respect to the cost function $C_i$. Hence, the set of feasible stationary policies for CMDP is then defined as $\Pi_C = \{\pi \in \Pi | \forall i, \mathcal{J}_{C_i}(\pi) \leq d_i\}$, where $d_i \in \mathbb{R}$. In CMDP, the objective is to select a feasible stationary policy $\pi$ that maximizes the performance: $\max_{\pi \in \Pi_\theta \cap \Pi_C} \mathcal{J}(\pi)$. Lastly, we define on-policy value, action-value, and advantage functions for the cost as $V_{C_i}^\pi$, $Q_{C_i}^\pi$ and $A_{C_i}^\pi$, which as analogous to $V^\pi$, $Q^\pi$, and $A^\pi$, with $C_i$ replacing $R$.

## 4 GUARD Safe RL Library

### 4.1 Overall Implementation

As a highly self contained safe RL benchmark, GUARD contains the latest methods that can achieve on-policy safe RL: (i) end-to-end safe RL algorithms including CPO (Achiam et al., 2017), TRPO-Lagrangian (Bohez et al., 2019), TRPO-FAC (Ma et al., 2021), TRPO-IPO (Liu et al., 2020), and PCPO (Yang et al., 2020b); (ii) hierarchical safe RL algorithms including TRPO-SL (TRPO-Safety Layer) (Dalal et al., 2018) and TRPO-USL (TRPO-Unrolling Safety Layer) (Zhang et al., 2022a). We also include TRPO (Schulman et al., 2015) as an unconstrained RL baseline. Note that GUARD only considers model-free approaches which rely less on assumptions than model-based ones. We highlight the benefits of our algorithm implementations in GUARD:

- GUARD comprehensively covers a **wide range of on-policy algorithms** that enforce safety in both hierarchical and end-to-end structures. Hierarchical methods maintain a separate safety layer, while end-to-end methods solve the constrained learning problem as a whole.

- GUARD provides a **fair comparison among safety components** by equipping every algorithm with the same reward-oriented RL backbone (i.e., TRPO (Schulman et al., 2015)), implementation (i.e., MLP policies with [64, 64] hidden layers and tanh activation), and training procedures. Hence, all algorithms inherit the performance guarantee of TRPO.

- GUARD is implemented in PyTorch with a clean structure where every algorithm is self-contained, enabling **fast customization and development** of new safe RL algorithms. GUARD also comes with unified logging and plotting utilities which makes analysis easy.

### 4.2 Unconstrained RL

**TRPO**  We include TRPO (Schulman et al., 2015) since it is state-of-the-art and several safe RL algorithms are based on it. TRPO is an unconstrained RL algorithm and only maximizes performance $\mathcal{J}$. The key idea behind TRPO is to iteratively update the policy within a local range (trust region) of the most recent version $\pi_k$. Mathematically, TRPO updates policy via

$$\pi_{k+1} = \arg\max_{\pi \in \Pi_\theta} \mathcal{J}(\pi) \qquad \textbf{s.t.} \, \mathcal{D}_{KL}(\pi, \pi_k) \leq \delta, \tag{2}$$

where $\mathcal{D}_{KL}$ is Kullback-Leibler (KL) divergence, $\delta > 0$ and the set $\{\pi \in \Pi_\theta \; : \; \mathcal{D}_{KL}(\pi, \pi_k) \leq \delta\}$ is called the *trust region*. To solve equation 2, TRPO applies Taylor expansion to the objective and constraint at $\pi_k$ to the first and second order, respectively. That results in an approximate optimization with linear objective and quadratic constraints (LOQC). TRPO guarantees a worst-case performance degradation.

### 4.3 End-to-End Safe RL

#### 4.3.1 Constrained Policy Optimization-based Algorithms

**CPO** Constrained Policy Optimizaiton (CPO) (Achiam et al., 2017) handles CMDP by extending TRPO. Similar to TRPO, CPO also performs local policy updates in a trust region. Different from TRPO, CPO additionally requires $\pi_{k+1}$ to be constrained by $\Pi_\theta \cap \Pi_C$. For practical implementation, CPO replaces the objective and constraints with surrogate functions (advantage functions), which can easily be estimated from samples collected on $\pi_k$, formally:

$$\pi_{k+1} = \arg\max_{\pi \in \Pi_\theta} \mathop{\mathbb{E}}_{\substack{s \sim d^{\pi_k} \\ a \sim \pi}} [A^{\pi_k}(s, a)] \tag{3}$$

$$\text{s.t.} \quad \mathcal{D}_{KL}(\pi, \pi_k) \leq \delta, \quad \mathcal{J}_{C_i}(\pi_k) + \frac{1}{1-\gamma} \mathop{\mathbb{E}}_{\substack{s \sim d^{\pi_k} \\ a \sim \pi}} \left[ A^{\pi_k}_{C_i}(s, a) \right] \leq d_i, i = 1, \cdots, m.$$

where $d^{\pi_k} \doteq (1-\gamma) \sum_{t=0}^{H} \gamma^t P(s_t = s|\pi_k)$ is the discounted state distribution. Following TRPO, CPO also performs Taylor expansion on the objective and constraints, resulting in a Linear Objective with Linear and Quadratic Constraints (LOLQC). CPO inherits the worst-case performance degradation guarantee from TRPO and has a worst-case cost violation guarantee.

**PCPO** Projection-based Constrained Policy Optimization (PCPO) (Yang et al., 2020b) is proposed based on CPO, where PCPO first maximizes reward using a trust region optimization method without any constraints, then PCPO reconciles the constraint violation (if any) by projecting the policy back onto the constraint set. Policy update then follows an analytical solution:

$$\pi_{k+1} = \pi_k + \sqrt{\frac{2\delta}{g^\top H^{-1} g}} H^{-1} g - \max\left( 0, \frac{\sqrt{\frac{2\delta}{g^\top H^{-1} g}} g_c^\top H^{-1} g + b}{g_c^\top L^{-1} g_c} \right) L^{-1} g_c \tag{4}$$

where $g_c$ is the gradient of the cost advantage function, $g$ is the gradient of the reward advantage function, $H$ is the Hessian of the KL divergence constraint, $b$ is the constraint violation of the policy $\pi_k$, $L = \mathcal{I}$ for $L_2$ norm projection, and $L = H$ for KL divergence projection. PCPO provides a lower bound on reward improvement and an upper bound on constraint violation.

#### 4.3.2 Lagrangian-based Algorithms

**TRPO-Lagrangian** Lagrangian methods solve constrained optimization by transforming hard constraints into soft constraints in the form of penalties for violations. Given the objective $\mathcal{J}(\pi)$ and constraints $\{\mathcal{J}_{C_i}(\pi) \leq d_i\}_i$, TRPO-Lagrangian (Bohez et al., 2019) first constructs the dual problem

$$\max_{\forall i, \lambda_i \geq 0} \min_{\pi \in \Pi_\theta} -\mathcal{J}(\pi) + \sum_i \lambda_i (\mathcal{J}_{C_i}(\pi) - d_i). \tag{5}$$

The update of $\theta$ is done via a trust region update with the objective of equation 2 replaced by that of equation 5 while fixing $\lambda_i$. The update of $\lambda_i$ is done via standard gradient ascend. Note that TRPO-Lagrangian does not have a theoretical guarantee for constraint satisfaction.

**TRPO-FAC** Inspired by Lagrangian methods and aiming at enforcing state-wise constraints (e.g., preventing state from stepping into infeasible parts in the state space), Feasible Actor Critic (FAC) (Ma et al., 2021)

introduces a multiplier (dual variable) network. Via an alternative update procedure similar to that for equation 5, TRPO-FAC solves the *statewise* Lagrangian objective:

$$\max_{\forall i, \xi_i} \min_{\pi \in \Pi_\theta} -\mathcal{J}(\pi) + \sum_i \mathbb{E}_{s \sim d^{\pi_k}} \left[ \lambda_{\xi_i}(s)(\mathcal{J}_{C_i}(\pi) - d_i) \right], \tag{6}$$

where $\lambda_{\xi_i}(s)$ is a parameterized Lagrangian multiplier network and is parameterized by $\xi_i$ for the $i$-th constraint. Note that TRPO-FAC does not have a theoretical guarantee for constraint satisfaction.

**TRPO-IPO**  TRPO-IPO (Liu et al., 2020) incorporates constraints by augmenting the optimization objective in equation 2 with logarithmic barrier functions, inspired by the interior-point method (Boyd & Vandenberghe, 2004). Ideally, the augmented objective is $I(\mathcal{J}_{C_i}(\pi) - d_i) = 0$ if $\mathcal{J}_{C_i}(\pi) - d_i \leq 0$ or $-\infty$ otherwise. Intuitively, that enforces the constraints since the violation penalty would be $-\infty$. To make the objective differentiable, $I(\cdot)$ is approximated by $\phi(x) = \log(-x)/t$ where $t > 0$ is a hyperparameter. Then TRPO-IPO solves equation 2 with the objective replaced by $\mathcal{J}_{\text{IPO}}(\pi) = \mathcal{J}(\pi) + \sum_i \phi(\mathcal{J}_{C_i}(x) - d_i)$. TRPO-IPO does not have theoretical guarantees for constraint satisfaction.

### 4.4  Hierarchical Safe RL

**Safety Layer**  Safety Layer (Dalal et al., 2018), added on top of the original policy network, conducts a quadratic-programming-based constrained optimization to project reference action into the nearest safe action. Mathematically:

$$a_t^{safe} = \arg \min_a \frac{1}{2} \|a - a_t^{ref}\|^2 \quad \textbf{s.t.} \quad \forall i, \bar{g}_{\varphi_i}(s_t)^\top a + C_i(s_{t-1}, a_{t-1}, s_t) \leq d_i \tag{7}$$

where $a_t^{ref} \sim \pi_k(\cdot|s_t)$, and $\bar{g}_{\varphi_i}(s_t)^\top a_t + C_i(s_{t-1}, a_{t-1}, s_t) \approx C_i(s_t, a_t, s_{t+1})$ is a $\varphi$ parameterized linear model. If there's only one constraint, equation 7 has a closed-form solution.

**USL**  Unrolling Safety Layer (USL) (Zhang et al., 2022b) is proposed to project the reference action into safe action via gradient-based correction. Specifically, USL iteratively updates the learned $Q_C(s, a)$ function with the samples collected during training. With step size $\eta$ and normalization factor $\mathcal{Z}$, USL performs gradient descent as $a_t^{safe} = a_t^{ref} - \frac{\eta}{\mathcal{Z}} \cdot \frac{\partial}{\partial a_t^{ref}}[Q_C(s_t, a_t^{ref}) - d]$.

## 5  GUARD Testing Suite

### 5.1  Robot Options

In GUARD testing suite, the agent (in the form of a robot) perceives the world through sensors and interacts with the world through actuators. Robots are specified through MuJoCo XML files. The suite is equipped with **8** types of pre-made robots that we use in our benchmark environments as shown in Figure 1. The action space of the robots are continuous, and linearly scaled to [-1, +1].

**Swimmer** consist of three links and two joints. Each joint connects two links to form a linear chain. Swimmer can move around by applying **2** torques on the joints.

**Ant** is a quadrupedal robot composed of a torso and four legs. Each of the four legs has a hip joint and a knee joint; and can move around by applying **8** torques to the joints.

**Walker** is a bipedal robot that consists of four main parts - a torso, two thighs, two legs, and two feet. Different from the knee joints and the ankle joints, each of the hip joints has three hinges in the $x$, $y$ and $z$ coordinates to help turning. With the torso height fixed, Walker can move around by controlling **10** joint torques.

**Humanoid** is also a bipedal robot that has a torso with a pair of legs and arms. Each leg of Humanoid consists of two joints (no ankle joint). Since we mainly focus on the navigation ability of the robots in

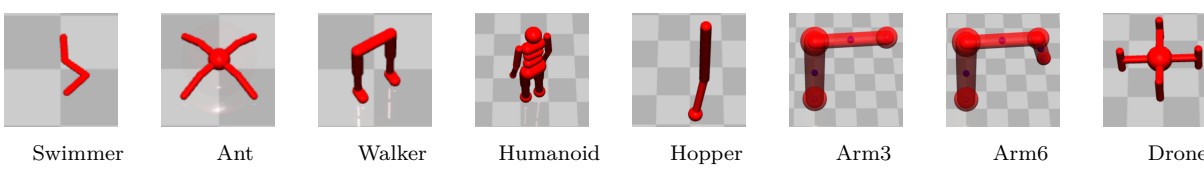

| Swimmer | Ant | Walker | Humanoid | Hopper | Arm3 | Arm6 | Drone |

Figure 1: Robots of our environments.

designed tasks, the arm joints of Humanoid are fixed, which enables Humanoid to move around by only controlling **6** torques.

**Hopper** is a one-legged robot that consists of four main parts - a torso, a thigh, a leg, and a single foot. Similar to Walker, Hopper can move around by controlling **5** joint torques.

**Arm3** is designed to simulate a fixed three-joint robot arm. Arm is equipped with multiple sensors on each link in order to fully observe the environment. By controlling **3** joint torques, Arm can move its end effector around with high flexibility.

**Arm6** is designed to simulate a robot manipulator with a fixed base and six joints. Similar to Arm3, Arm6 can move its end effector around by controlling **6** torques.

**Drone** is designed to simulate a quadrotor. The interaction between the quadrotor and the air is simulated by applying four external forces on each of the propellers. The external forces are set to balance the gravity when the control action is zero. Drone can move in 3D space by applying **4** additional control forces on the propellers.

## 5.2 Task Options

We categorize robot tasks in two ways: (i) interactive versus non-interactive tasks, and (ii) 2D space versus 3D space tasks. 2D space tasks constrain agents to a planar space, while 3D space tasks do not. Non-interactive tasks primarily involve achieving a target state (e.g., trajectory tracking) while interactive tasks (e.g., human-robot collaboration and unstructured object pickup) necessitate contact or non-contact interactions between the robot and humans or movable objects, rendering them more challenging. On a variety of tasks that cover different situations, GUARD facilitates a thorough evaluation of safe RL algorithms via the following tasks. See Table 17 for more information.

**Goal** (Figure 2a) requires the robot to navigate towards a series of 2D or 3D goal positions. Upon reaching a goal, the location is randomly reset. The task provides a sparse reward upon goal achievement and a dense reward for making progress toward the goal.

**Push** (Figure 2b) requires the robot pushing a ball toward different goal positions. The task includes a sparse reward for the ball reaching the goal circle and a dense reward that encourages the agent to approach both the ball and the goal. Unlike pushing a box in Safety Gym, it is more challenging to push a ball since the ball can roll away and the contact dynamics are more complex.

**Chase** (Figure 2c) requires the robot tracking multiple dynamic targets. Those targets continuously move away from the robot at a slow speed. The dense reward component provides a bonus for minimizing the distance between the robot and the targets. The targets are constrained to a circular area. A 3D version of this task is also available, where the targets move within a restricted 3D space. Detailed dynamics of the targets is described in Appendix B.5.1.

**Defense** (Figure 2d) requires the robot to prevent dynamic targets from entering a protected circle area. The targets will head straight toward the protected area or avoid the robot if the robot gets too close. Dense reward component provides a bonus for increasing the cumulative distance between the targets and the protected area. Detailed dynamics of the targets is described in Appendix B.5.2.

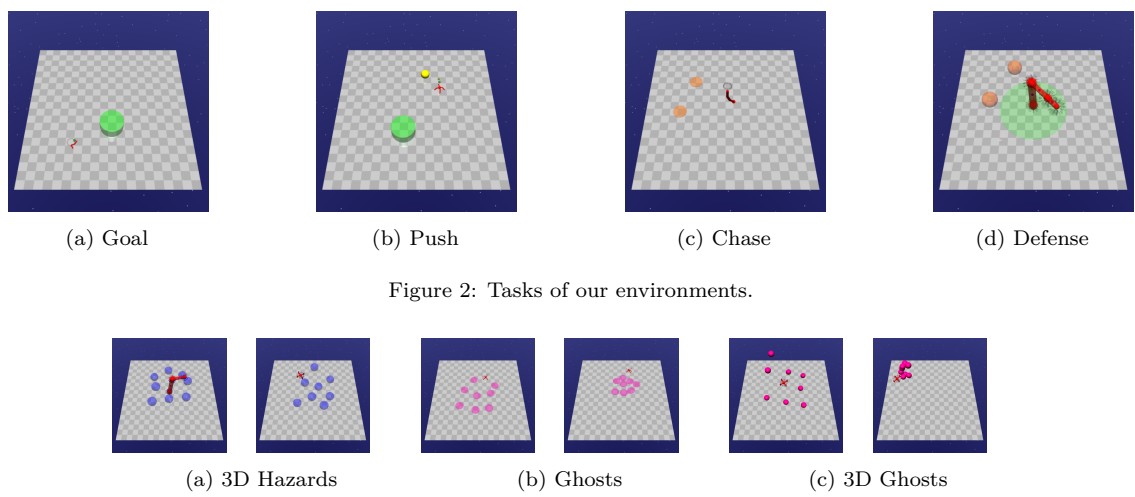

Figure 2: Tasks of our environments.

(a) Goal     (b) Push     (c) Chase     (d) Defense

(a) 3D Hazards     (b) Ghosts     (c) 3D Ghosts

Figure 3: Constraints of our environments.

## 5.3 Constraint Options

We classify constraints based on various factors: **trespassibility**: whether constraints are trespassable or non-trespassable. Trespassable constraints allow violations without causing any changes to the robot's behaviors, and vice versa. (ii) **movability**: whether they are immovable, passively movable, or actively movable; and (iii) **motion space**: whether they pertain to 2D or 3D environments. To cover a comprehensive range of constraint configurations, we introduce additional constraint types via expanding Safety Gym. Please refer to Table 18 for all configurable constraints.

**3D Hazards** (Figure 3a) are dangerous 3D areas to avoid. These are floating spheres that are trespassable, and the robot is penalized for entering them.

**Ghosts** (Figure 3b) are dangerous areas to avoid. Different from hazards, ghosts always move toward the robot slowly, represented by circles on the ground. Ghosts can be either trespassable or non-trespassable. The robot is penalized for touching the non-trespassable ghosts and entering the trespassable ghosts. Moreover, ghosts can be configured to start chasing the robot when the distance from the robot is larger than some threshold. This feature together with the adjustable velocity allows users to design the ghosts with different aggressiveness. Detailed dynamics of the targets is described in Appendix B.5.3.

**3D Ghosts** (Figure 3c) are dangerous 3D areas to avoid. These are floating spheres as 3D versions of ghosts, sharing the similar behavior with ghosts.

## 6 GUARD Experiments

In our experiments, we aim to answer these questions:

**Q1** What are the overall benchmark results?

**Q2** How does the difficulty of constraints impact the algorithm performance?

**Q3** What is the detailed performance of different categories of safe RL algorithms?

**Q4** How does task complexity impact algorithm performance?

**Q5** How does adaptive multiplier impact Lagrangian-based methods?

**Q6** How does feasibility projection impact CPO-based methods?

**Q7** How does cost dynamics linearization impact Hierarchical-based methods?

**Q8** What is the individual algorithm performance across all tasks?

## 6.1 Experiment Setup

In GUARD experiments, our objective is to assess the performance of safe RL algorithms across a diverse range of benchmark testing suites. These suites are meticulously designed, incorporating all available robot options as detailed in Section 5.1 and all task options outlined in Section 5.2. Additionally, we offer seamless integration of various constraint options into these benchmark testing suites, allowing users to select desired constraint types, numbers, sizes, and other parameters. Considering the diversity in robots, tasks, constraint types, and difficulty levels, we have curated 72 test suites. These predefined benchmark testing suites follow the format `{Task}_{Robot}_{Constraint Number}{Constraint Type}`. For a comprehensive list of our testing suites, please refer to Table 20.

**Remark 1.** *Note that the ladder of constraint difficulty levels can be readily established by introducing varying numbers, sizes, and types of constraints, as detailed in [Section 4.2, (Ray et al., 2019a)].*

**Comparison Group** The methods in the comparison group include all methods in GUARD Safe RL Libraray: (i) unconstrained RL algorithm TRPO (Schulman et al., 2015) (ii) end-to-end constrained safe RL algorithms CPO (Achiam et al., 2017), TRPO-Lagrangian (Bohez et al., 2019), TRPO-FAC (Ma et al., 2021), TRPO-IPO (Liu et al., 2020), PCPO (Yang et al., 2020b), and (iii) hierarchical safe RL algorithms TRPO-SL (TRPO-Safety Layer) (Dalal et al., 2018), TRPO-USL (TRPO-Unrolling Safety Layer) (Zhang et al., 2022a). For hierarchical safe RL algorithms, we incorporate a warm-up phase (constituting 1/3 of the total epochs) dedicated to unconstrained

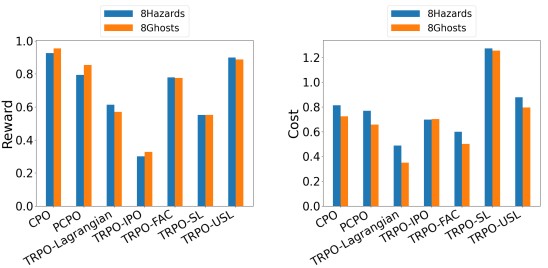

Figure 4: constraint difficulty ablation study with Goal_Point_8{Constraint}

TRPO training. The data generated during this phase is utilized to pre-train the safety critic for subsequent epochs. The target cost for all safe RL methods is set to zero, aligning with our objective of achieving zero violations. To ensure consistency, shared configurations, including hidden layers, learning rate, and target KL, are uniformly applied across all methods. Simultaneously, unique parameters such as Lagrangian learning rate, IPO parameter, and Warmup ratio are fine-tuned to optimize the performance of each respective method. Further details are provided in Table 19.

## 6.2 Evaluating GUARD Safe RL Library and Comparison Analysis

**Overall Benchmark Results** The summarized results can be found in Tables 21 to 25, and the learning rate curves are presented in Figures 14 to 18. In Figure 5, we select 8 sets of results to demonstrate the performance of different robots, tasks and constraints in GUARD. At a high level, the experiments show that all methods can consistently improve reward performance.

Different methods have different trade-offs between rewards and cost. When comparing constrained RL methods to unconstrained RL methods, the former exhibit superior performance in terms of cost reduction. By incorporating constraints into the RL framework, the robot can navigate its environment while minimizing costs. This feature is particularly crucial for real-world applications where the avoidance of hazards and obstacles is of utmost importance. However, current safe RL algorithms (i.e. CPO, PCPO and Lagrangian methods) are hard to achieve zero-violation performance even when the cost threshold is set as zero. Compared with them, hierarchical RL methods (i.e., TRPO-SL and TRPO-USL) can perform better at cost reduction. Nevertheless, although these methods excel at minimizing costs, they may sacrifice some degree of reward attainment in the process.

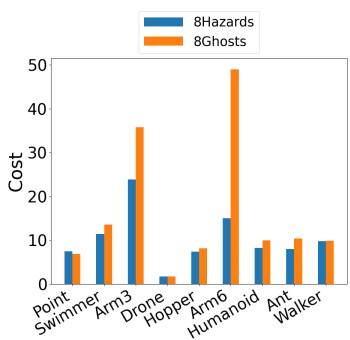

Figure 6: Raw TRPO cost score in Goal_{Robot}_8{Constraint} with different constraint difficulty.

**Impacts of Constraints Difficulty** To gain an intuitive understanding of constraint difficulty, we present a visual comparison in Figure 6. The

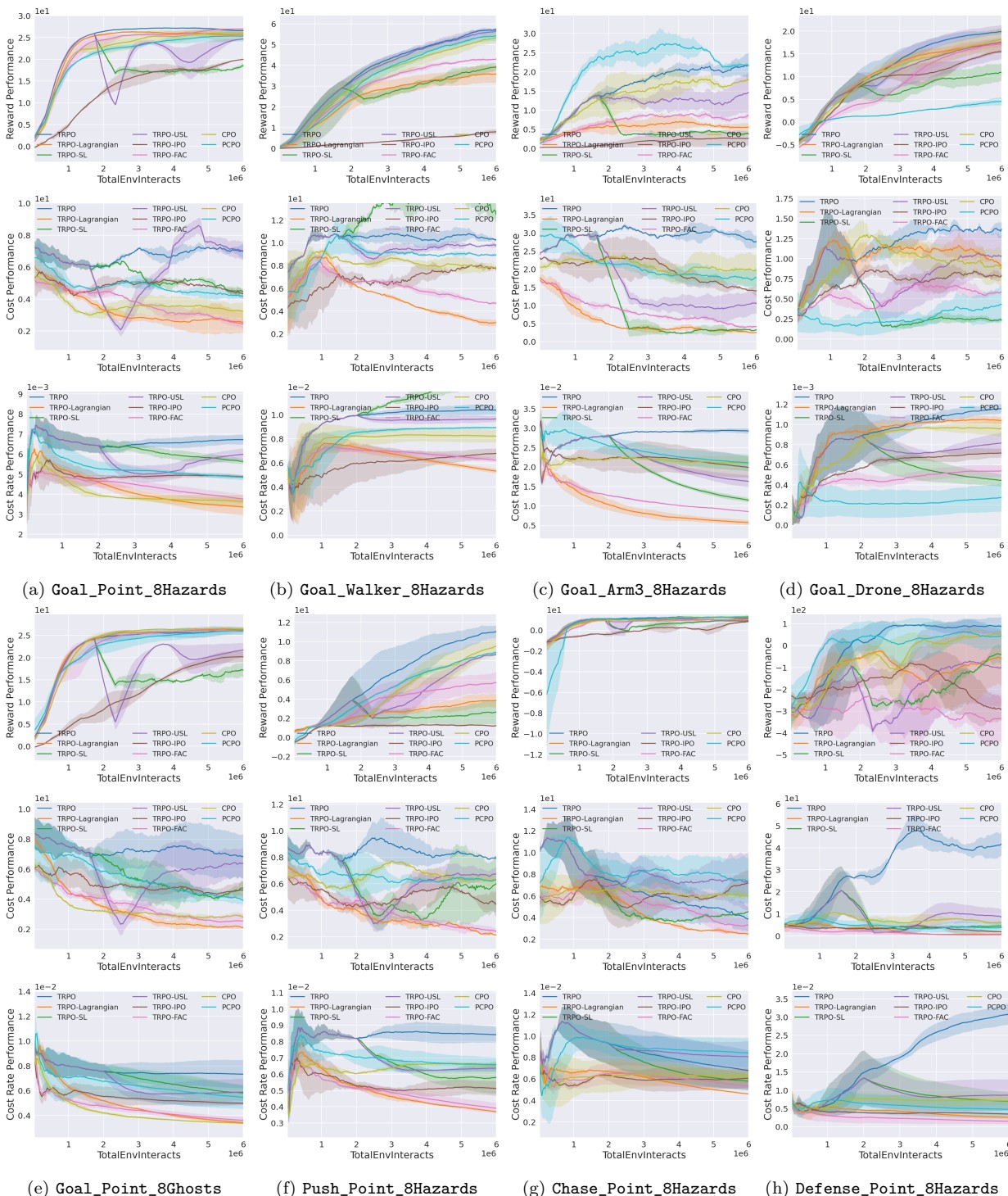

Figure 5: Comparison of results from four representative tasks. (a) to (d) cover four robots on the goal task. (e) shows the performance of a task with ghosts. (f) to (h) cover three different tasks with the point robot.

figure illustrates the raw cost comparison for TRPO across different robot options on the testing suite `Goal_{Robot}_8{Constraint}`, where we vary `8{Constraint}` from `8Hazards` to `8Ghosts`. It is essential to note that Hazards and Ghosts share the same cost computation rule, with the distinction that Ghosts adversarially move towards the robot. The comparison reveals some significant increase in raw TRPO cost when transitioning from `8Hazards` to `8Ghosts` for every robot option, indicating the escalating challenge posed to safe RL algorithms.

To assess the impact of constraint difficulty on the performance of various safe RL algorithms, we analyze the performance changes across different algorithms using the `Goal_Point_8{Constraint}` testing suites. Specifically, we vary `8{Constraint}` from `8Hazards` to `8Ghosts`, with the former representing an easier constraint where all hazards are static, and the latter indicating a more challenging constraint where all hazards are adversarially moving towards the robot. To ensure a fair comparison across diverse tasks, we report the normalized reward and normalized cost for each algorithm using the following metric:

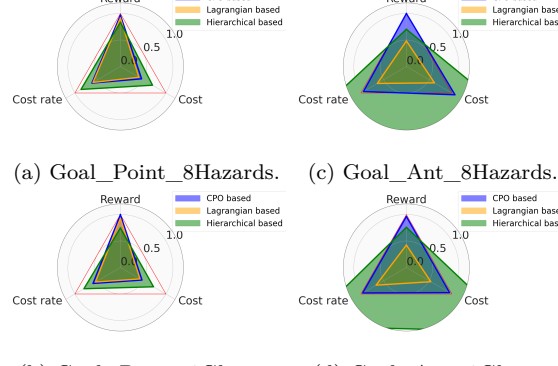

(a) Goal__Point__8Hazards.  (c) Goal__Ant__8Hazards.

(b) Goal__Point__8Ghosts.  (d) Goal__Ant__8Ghosts.

$$reward_{normalized} = \min\left(\frac{reward_{algorithm}}{reward_{TRPO}}, 1\right) \quad (8)$$

$$cost_{normalized} = \frac{cost_{algorithm}}{cost_{TRPO}} \quad (9)$$

where $reward_{normalized}$ are obtained from Tables 21 to 25. Before applying Equation (8), in cases where negative values exist for $reward_{normalized}$ in the results, we shift all values to be above zero with respect to the minimum $reward_{normalized}$ observed within the same experiment.

Figure 7: Comparison of performance of different categories of algorithms in four representative test suites across low-to-high dimensional robots and easy-to-hard constraints. The scores of reward, cost and cost rate are normalized with respect to TRPO. Red triangle serves as the baseline TRPO performance. A higher normalized reward score and lower normalized cost/cost rate scores indicate better performance along each axis.

The comparative results are succinctly presented in Figure 4. Notably, as the constraint difficulty transitions from an easier to a more challenging level, majority algorithms within the Safe RL Library showcase safer policy learning behaviors. Additionally, they manage to maintain roughly the same, and in some instances, achieve higher reward performance. This observation underscores two key findings: (i) safe RL algorithms exhibit resilience to the increased difficulty of constraints, and (ii) heightened constraint difficulty serves as a more effective metric for distinguishing safety performance. Notably, the cost performance gap between TRPO-Lagrangian and TRPO-IPO is magnified in the Ghost environment, where a more sophisticated safe policy is in need to avoid adversarial hazards.

**Characteristics of Different Categories of Safe RL Algorithms** Within the Safe RL Library, three primary categories of safe RL algorithms exist: (i) Lagrangian-based methods (TRPO-Lagrangian, TRPO-FAC, TRPO-IPO), (ii) constrained policy optimization-based methods (CPO, PCPO), and (iii) hierarchical-based methods (Safe Layer, USL). To investigate the performance characteristics inherent in each category, we carefully select four representative testing suites spanning high/low dimensional robots and easy/hard constraints. Subsequently, we chart the algorithm-wise averaged TRPO normalized reward, cost, and cost rate. For example, the algorithm-wise averaged score for hierarchical-based methods is computed by taking the mean across Safe Layer and USL. Additionally, the TRPO normalized cost rate is defined as:

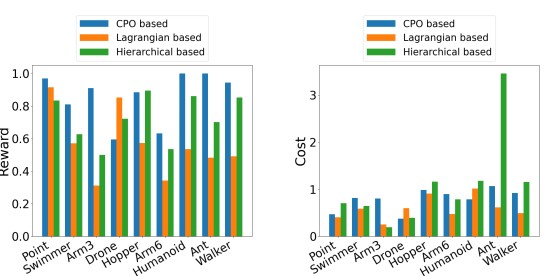

Figure 8: TRPO normalized performance metrics (Reward and Cost) of three algorithm categories across varied difficulty levels in Goal__{Robot}__8Hazards

$$cost\ rate_{normalized} = \frac{cost\ rate_{algorithm}}{cost\ rate_{TRPO}} \quad (10)$$

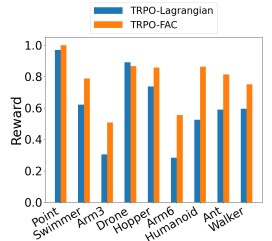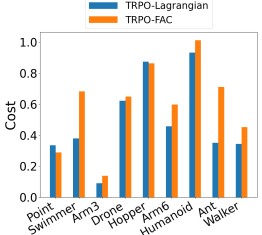

The summarized comparison results are presented in fig. 7. Generally, in terms of safety performance, Lagrangian-based methods demonstrate the highest efficacy, followed by CPO-based methods, while Hierarchical-based methods exhibit comparatively lower performance. Several key observations emerge: (i) Lagrangian-based methods prioritize safety performance and are willing to make concessions in reward performance when necessary, as evidenced by significantly lower reward performance in high-dimensional

Figure 9: TRPO normalized performance metrics (Reward and Cost) of Lagrangian and FAC across varied difficulty levels in Goal_{Robot}_8Hazards

tasks. This behavior of Lagrangian-based methods have also been witnessed in [Figure 9, Doggo experiments, (Ray et al., 2019a)] (ii) CPO-based methods tend to maintain a high standard of reward performance akin to unconstrained RL, but this strategy compromises safe behavior, particularly in high-dimensional tasks. This observation exactly matches the reported results [Figure 8, Figure 9 (Ray et al., 2019a)] [Figure 6, (Yang et al., 2023)]. (iii) Hierarchical-based methods only demonstrate effectiveness in low-dimensional systems, struggling to learn reasonable safe behavior in high-dimensional systems. This limitation is attributed to the escalating complexity of cost function dynamics in high-dimensional systems, posing challenges for effective cost function approximation. This finding aligns with the results reported in [Figure 3, Safety Layer, Recovery RL, (Zhang et al., 2023)][Section 6.1, (Zhao et al., 2021)].

**Effects of Task Complexity on Various Categories of Safe RL Algorithms** To examine the impact of increased task complexity, particularly in the context of higher degrees of freedom (DOF) in robots, we summarize the TRPO normalized performance metrics (reward and cost) for three algorithm categories across various difficulty levels in the Goal_{Robot}_8Hazards testing suites, with an incremental increase in robot DOF, in Figure 8. The performance characteristics of different algorithm categories exhibit interesting variations with heightened task complexity.

Surprisingly, there is no discernible trend in reward and cost performance across all algorithm categories as task complexity increases. However, Lagrangian and Hierarchical-based methods each align better with specific robot types. (i) For Arm robots, Lagrangian-based methods struggle to learn reward, whereas Hierarchical-based methods perform well on both reward and safety. (ii) In the case of linked/legged robots, Lagrangian-based methods maintain a consistent, albeit mediocre, reward performance, coupled with optimal safety performance. Conversely, Hierarchical-based methods struggle to learn a safe policy, particularly on the Ant robot.

Regarding CPO-based methods, their reward performance remains consistently high even with increased task complexity. However, although CPO-based methods exhibit effective learning of safe policies with low-dimensional robots, they encounter difficulties in learning optimal safe behavior as task complexity exceeds a certain threshold.

**Impacts of Adaptive Multiplier in Lagrangian-based Methods** An essential differentiator between TRPO-FAC and TRPO-Lagrangian lies in the incorporation of a multiplier network, i.e. an adaptive multiplier. To gain a comprehensive understanding of this technique, we compare the normalized performance metrics on Goal_{Robot}_8Hazards testing suites, both with and without this feature, as illustrated in Figure 9. It is evident that the introduction of the adaptive multiplier results in consistently stable and favorable performance across various difficulty levels of tasks. This reward behavior of adaptive multiplier has been observed in [Fig 2, (Ma et al., 2021)] However, as a trade-off, the adaptive multiplier tends to compromise safety performance, leading to higher costs, particularly evident in Swimmer and Ant. This cost behavior of adaptive multiplier has been reported in [Fig 3, FAC w/ $\phi_0$, (Ma et al., 2022)].

**Impacts of Feasibility Projection in CPO-based Methods** The policy rule of PCPO enhances CPO by projecting the reward-oriented policy back into the constraint set, ensuring a feasible policy update at every iteration (feasibility projection). To assess the impact of this technique, we conduct a comparison of normalized performance metrics on Goal_{Robot}_8Hazards testing suites, both with and without this feature, as depicted in Figure 10. The figure indicates that feasibility projection is potent in trading off performance for significantly safer behavior in specific tasks, such as the Drone task. This behavior

of feasibility projection has been observed in [Fig 4(e), (Yang et al., 2020a)]. However, this may not be universally applicable, as feasibility projection often leads to slightly less safe behavior and, in some cases, adversely affects reward performance without corresponding improvements in safety, as observed in the Arm6 scenario.

**Impacts of Linearized Cost Dynamics** Hierarchical methods offer the explicit projection of an unsafe reference action into a safe action set, a process that involves determining the cost given a state-action pair. While one approach involves directly learning the cost function, as seen in USL, SafeLayer streamlines the process by linearizing the cost function dynamics and focusing solely on learning the gradient. To assess the effectiveness of this method, we conduct a comparison of normalized performance metrics on Goal_{Robot}_8Hazards testing suites, both with and without this feature, as depicted in Figure 11. The results reveal that the linearization is highly effective in achieving robust safe behavior in low-dimensional robot

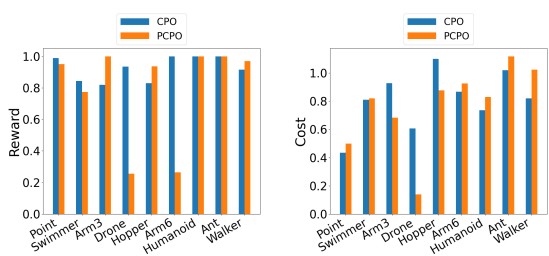

Figure 10: TRPO normalized performance metrics (Reward and Cost) of CPO and PCPO across varied difficulty levels in Goal_{Robot}_8Hazards

tasks, particularly evident in the cases of Drone and Arm3. However, the linear approximation of the cost function in SafeLayer becomes less accurate in scenarios with highly nonlinear dynamics, such as the complex Ant robot, leading to a significant increase in overall costs. Therefore, Linearization emerges as a powerful technique for tasks with low complexity, especially those involving low-dimensional robots. The similar failure of cost dynamics linearization on complex tasks has been observed in [Fig 6 (c), Fig 6 (d), (Zhao et al., 2021)].

**Comprehensive Evaluation of Algorithm Performance Across All Tasks** Lastly, for a holistic understanding of the performance exhibited by each algorithm within the Safe RL Library across all tasks within the GUARD Testing suites, we present spider plots in Figure 12. These plots illustrate the TRPO normalized reward, cost, and cost rate for each algorithm across five task options, with all performance metrics averaged over all available robot options. TRPO is utilized as the baseline in this figure, emphasizing its focus on maximizing reward performance.

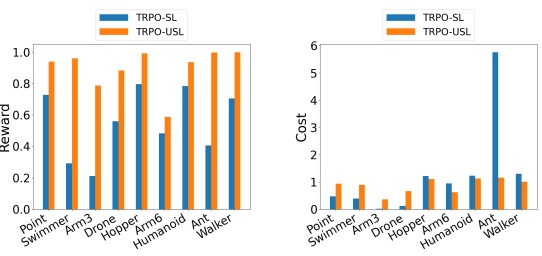

Figure 11: TRPO normalized performance metrics (Reward and Cost) of USL and SafeLayer across varied difficulty levels in Goal_{Robot}_8Hazards

Within the category of CPO-based methods: (i) CPO maintains reward levels comparable to TRPO across all tasks but struggles to learn a safe policy in the Chase task. (ii) PCPO exhibits a lower performance in both reward and safety aspects, particularly in Chase and Defense tasks.

Within the category of Lagrangian-based methods: (i) Lagrangian maintains a balance between good reward and cost across all tasks, although its reward performance is not exceptional. (ii) FAC excels in achieving higher reward across all tasks, with a slight sacrifice in safety performance in all tasks except Defense. (iii) IPO performs the worst in both reward and safety metrics, although it still manages to learn viable safe policies for all tasks.

Within the category of Hierarchical-based methods: (i) USL performs admirably in achieving satisfactory rewards for all tasks but only learns mediocre safe policies and struggles with the Chase task. (ii) SafeLayer demonstrates poor performance in reward performance and consistently fails to learn safe policies in most tasks. However, it excels in achieving satisfactory safety performance, particularly in the Defense task.

## 7    Conclusions

This paper introduces GUARD, the **G**eneralized **U**nified **SA**fe **R**einforcement Learning **D**evelopment Benchmark. GUARD offers several advantages over existing benchmarks. Firstly, it provides a generalized framework

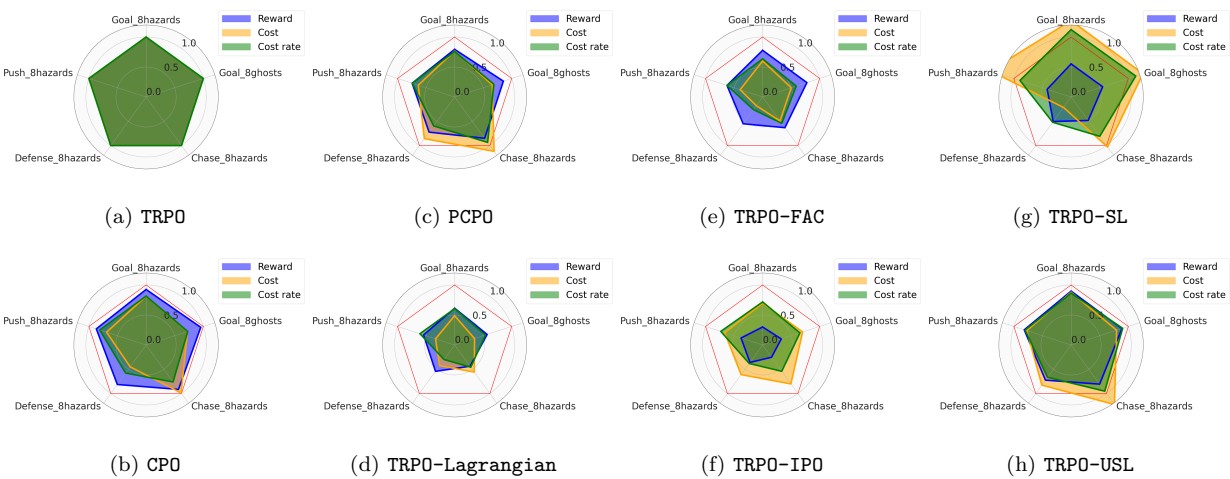

Figure 12: Algorithm performance over different tasks. The scores of reward, cost and cost rate are normalized with respect to TRPO. The score for each task is averaged over all robots. Red boundary serves as the baseline TRPO performance. A higher normalized reward score and lower normalized cost/cost rate scores indicate better performance along each axis.

with a wide range of RL agents, tasks, and constraint specifications. Secondly, GUARD has self-contained implementations of a comprehensive range of state-of-the-art safe RL algorithms. Lastly, GUARD is highly customizable, allowing researchers to tailor tasks and algorithms to specific needs. Using GUARD, we present a comparative analysis of state-of-the-art safe RL algorithms across various task settings, establishing essential baselines for future research.

**Future work**   In our future endeavors, we aim to work on the following expansions: (i) To enhance the ease of deploying Reinforcement Learning (RL) methods on real robots, we are actively incorporating additional realistic robot models into GUARD. (ii) The current GUARD tasks primarily focus on robot locomotion; however, we are planning to broaden the spectrum of available task types to empower users with a more extensive testing suite, including options such as speed control and contact-rich safety. (iii) Lastly, we plan to extend the range of safe RL libraries to include the latest algorithms such as off-policy methods.

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

## A  Appendix

## B  Environment Details

### B.1  Observation Space and Action space of different robots

The action space and observation space of different robots are summarized in Tables 1 to 16

Table 1: Action space of Swimmer

| Num | Action | Min | Max | Name in XML | Joint | Unit |
|---|---|---|---|---|---|---|
| 0 | Torque applied on the first rotor | -1 | 1 | motor1_rot | hinge | torque $(Nm)$ |
| 1 | Torque applied on the second rotor | -1 | 1 | motor2_rot | hinge | torque $(Nm)$ |

Table 2: Observation space of Swimmer

| Num | Observation | Min | Max | Name in XML | Joint | Unit |
|---|---|---|---|---|---|---|
| 0/1/2 | 3-axis linear acceleration of the torso (including gravity) | -Inf | Inf | accelerometer | free | acceleration $(m/s^2)$ |
| 3/4/5 | 3-axis linear velocity of the torso | -Inf | Inf | velocimeter | free | velocity $(m/s)$ |
| 6/7/8 | 3-axis angular velocity velocity of the torso | -Inf | Inf | gyro | free | angular velocity $(rad/s)$ |
| 9/10/11 | 3D magnetic flux vector at the torso | -Inf | Inf | magnetometer | free | magnetic flux density $(T)$ |
| 12 | Contact force at the first tip | 0 | Inf | touch_point1 | - | force$(N)$ |
| 13 | Contact force at the first rotor | 0 | Inf | touch_point2 | - | force$(N)$ |
| 14 | Contact force at the second rotor | 0 | Inf | touch_point3 | - | force$(N)$ |
| 15 | Contact force at the second tip | 0 | Inf | touch_point4 | - | force$(N)$ |
| 16 | Angle of the first rotor | -Inf | Inf | jointpos_motor1_rot | hinge | angle $(rad)$ |
| 17 | Angle of the second rotor | -Inf | Inf | jointpos_motor2_rot | hinge | angle $(rad)$ |
| 18 | Angular velocity of the first rotor | -Inf | Inf | jointvel_motor1_rot | hinge | angular velocity $(rad/s)$ |
| 19 | Angular velocity of the second rotor | -Inf | Inf | jointvel_motor2_rot | hinge | angular velocity $(rad/s)$ |

Table 3: Action space of Ant

| Num | Action | Min | Max | Name in XML | Joint | Unit |
|---|---|---|---|---|---|---|
| 0 | Torque applied on the rotor between the torso and front left hip | -1 | 1 | hip_1 | hinge | torque ($Nm$) |
| 1 | Torque applied on the rotor between the front left two links | -1 | 1 | ankle_1 | hinge | torque ($Nm$) |
| 2 | Torque applied on the rotor between the torso and front right hip | -1 | 1 | hip_2 | hinge | torque ($Nm$) |
| 3 | Torque applied on the rotor between the front right two links | -1 | 1 | ankle_2 | hinge | torque ($Nm$) |
| 4 | Torque applied on the rotor between the torso and back left hip | -1 | 1 | hip_3 | hinge | torque ($Nm$) |
| 5 | Torque applied on the rotor between the back left two links | -1 | 1 | ankle_3 | hinge | torque ($Nm$) |
| 6 | Torque applied on the rotor between the torso and back right hip | -1 | 1 | hip_4 | hinge | torque ($Nm$) |
| 7 | Torque applied on the rotor between the back right two links | -1 | 1 | ankle_4 | hinge | torque ($Nm$) |

Table 4: Observation space of Ant

| Num | Observation | Min | Max | Name in XML | Joint | Unit |
|---|---|---|---|---|---|---|
| 0/1/2 | 3-axis linear acceleration of the torso (including gravity) | -Inf | Inf | accelerometer | free | acceleration ($m/s^2$) |
| 3/4/5 | 3-axis linear velocity of the torso | -Inf | Inf | velocimeter | free | velocity ($m/s$) |
| 6/7/8 | 3-axis angular velocity velocity of the torso | -Inf | Inf | gyro | free | angular velocity ($rad/s$) |
| 9/10/11 | 3D magnetic flux vector at the torso | -Inf | Inf | magnetometer | free | magnetic flux density ($T$) |
| 12 | Contact force at the front left ankle | 0 | Inf | touch_ankle_1a | - | force($N$) |
| 13 | Contact force at the front right ankle | 0 | Inf | touch_ankle_2a | - | force($N$) |
| 14 | Contact force at the back left ankle | 0 | Inf | touch_ankle_3a | - | force($N$) |
| 15 | Contact force at the back left ankle | 0 | Inf | touch_ankle_4a | - | force($N$) |
| 16 | Contact force at the end of the front left leg | 0 | Inf | touch_ankle_1b | - | force($N$) |
| 17 | Contact force at the end of the front right leg | 0 | Inf | touch_ankle_2b | - | force($N$) |
| 18 | Contact force at the end of the back left leg | 0 | Inf | touch_ankle_3b | - | force($N$) |
| 19 | Contact force at the end of the back left leg | 0 | Inf | touch_ankle_4b | - | force($N$) |
| 20 | Angle of the front left hip | -Inf | Inf | jointpos_hip_1 | hinge | angle ($rad$) |
| 21 | Angle of the front right hip | -Inf | Inf | jointpos_hip_2 | hinge | angle ($rad$) |
| 22 | Angle of the back left hip | -Inf | Inf | jointpos_hip_3 | hinge | angle ($rad$) |
| 23 | Angle of the back right hip | -Inf | Inf | jointpos_hip_4 | hinge | angle ($rad$) |
| 24 | Angle of the front left ankle | -Inf | Inf | jointpos_ankle_1 | hinge | angle ($rad$) |
| 25 | Angle of the front right ankle | -Inf | Inf | jointpos_ankle_2 | hinge | angle ($rad$) |
| 26 | Angle of the back left ankle | -Inf | Inf | jointpos_ankle_3 | hinge | angle ($rad$) |
| 27 | Angle of the back right ankle | -Inf | Inf | jointpos_ankle_4 | hinge | angle ($rad$) |
| 28 | Angular velocity of the front left hip | -Inf | Inf | jointvel_hip_1 | hinge | angular velocity ($rad/s$) |
| 29 | Angular velocity of the front right hip | -Inf | Inf | jointvel_hip_2 | hinge | angular velocity ($rad/s$) |
| 30 | Angular velocity of the back left hip | -Inf | Inf | jointvel_hip_3 | hinge | angular velocity ($rad/s$) |
| 31 | Angular velocity of the back right hip | -Inf | Inf | jointvel_hip_4 | hinge | angular velocity ($rad/s$) |
| 32 | Angular velocity of the front left ankle | -Inf | Inf | jointvel_ankle_1 | hinge | angular velocity ($rad/s$) |
| 33 | Angular velocity of the front right ankle | -Inf | Inf | jointvel_ankle_2 | hinge | angular velocity ($rad/s$) |
| 34 | Angular velocity of the back left ankle | -Inf | Inf | jointvel_ankle_3 | hinge | angular velocity ($rad/s$) |
| 35 | Angular velocity of the back right ankle | -Inf | Inf | jointvel_ankle_4 | hinge | angular velocity ($rad/s$) |

Table 5: Action space of Walker

| Num | Action | Min | Max | Name in XML | Joint | Unit |
| --- | --- | --- | --- | --- | --- | --- |
| 0 | Torque applied on the rotor between torso and the right hip (x-coordinate) | -1 | 1 | right_hip_x | hinge | torque ($Nm$) |
| 1 | Torque applied on the rotor between torso and the right hip (z-coordinate) | -1 | 1 | right_hip_z | hinge | torque ($Nm$) |
| 2 | Torque applied on the rotor between torso and the right hip (y-coordinate) | -1 | 1 | right_hip_y | hinge | torque ($Nm$) |
| 3 | Torque applied on the right leg rotor | -1 | 1 | right_leg_joint | hinge | torque ($Nm$) |
| 4 | Torque applied on the right foot rotor | -1 | 1 | right_foot_joint | hinge | torque ($Nm$) |
| 5 | Torque applied on the rotor between torso and the left hip (x-coordinate) | -1 | 1 | left_hip_x | hinge | torque ($Nm$) |
| 6 | Torque applied on the rotor between torso and the left hip (z-coordinate) | -1 | 1 | left_hip_z | hinge | torque ($Nm$) |
| 7 | Torque applied on the rotor between torso and the left hip (y-coordinate) | -1 | 1 | left_hip_y | hinge | torque ($Nm$) |
| 8 | Torque applied on the left leg rotor | -1 | 1 | left_leg_joint | hinge | torque ($Nm$) |
| 9 | Torque applied on the left foot rotor | -1 | 1 | left_foot_joint | hinge | torque ($Nm$) |

Table 6: Observation space of Walker

| Num | Observation | Min | Max | Name in XML | Joint | Unit |
| --- | --- | --- | --- | --- | --- | --- |
| 0/1/2 | 3-axis linear acceleration of the torso (including gravity) | -Inf | Inf | accelerometer | free | acceleration ($m/s^2$) |
| 3/4/5 | 3-axis linear velocity of the torso | -Inf | Inf | velocimeter | free | velocity ($m/s$) |
| 6/7/8 | 3-axis angular velocity of the torso | -Inf | Inf | gyro | free | angular velocity ($rad/s$) |
| 9/10/11 | 3D magnetic flux vector at the torso | -Inf | Inf | magnetometer | free | magnetic flux density ($T$) |
| 12 | Contact force at the right foot | 0 | Inf | touch_right_foot | - | force($N$) |
| 13 | Contact force at the left foot | 0 | Inf | touch_left_foot | - | force($N$) |
| 14 | Angle of the right hip (x-coordinate) | -Inf | Inf | jointpos_right_hip_x | hinge | angle ($rad$) |
| 15 | Angle of the right hip (z-coordinate) | -Inf | Inf | jointpos_right_hip_z | hinge | angle ($rad$) |
| 16 | Angle of the right hip (y-coordinate) | -Inf | Inf | jointpos_right_hip_y | hinge | angle ($rad$) |
| 17 | Angle of the right leg | -Inf | Inf | jointpos_right_leg | hinge | angle ($rad$) |
| 18 | Angle of the right foot | -Inf | Inf | jointpos_right_foot | hinge | angle ($rad$) |
| 19 | Angle of the left hip (x-coordinate) | -Inf | Inf | jointpos_left_hip_x | hinge | angle ($rad$) |
| 20 | Angle of the left hip (z-coordinate) | -Inf | Inf | jointpos_left_hip_z | hinge | angle ($rad$) |
| 21 | Angle of the left hip (y-coordinate) | -Inf | Inf | jointpos_left_hip_y | hinge | angle ($rad$) |
| 22 | Angle of the left leg | -Inf | Inf | jointpos_left_leg | hinge | angle ($rad$) |
| 23 | Angle of the left foot | -Inf | Inf | jointpos_left_foot | hinge | angle ($rad$) |
| 24 | Angular velocity of the right hip (x-coordinate) | -Inf | Inf | jointvel_right_hip_x | hinge | angular velocity ($rad/s$) |
| 25 | Angular velocity of the right hip (z-coordinate) | -Inf | Inf | jointvel_right_hip_z | hinge | angular velocity ($rad/s$) |
| 26 | Angular velocity of the right hip (y-coordinate) | -Inf | Inf | jointvel_right_hip_y | hinge | angular velocity ($rad/s$) |
| 27 | Angular velocity of the right leg | -Inf | Inf | jointvel_right_leg | hinge | angular velocity ($rad/s$) |
| 28 | Angular velocity of the right foot | -Inf | Inf | jointvel_right_foot | hinge | angular velocity ($rad/s$) |
| 29 | Angular velocity of the left hip (x-coordinate) | -Inf | Inf | jointvel_left_hip_x | hinge | angular velocity ($rad/s$) |
| 30 | Angular velocity of the left hip (z-coordinate) | -Inf | Inf | jointvel_left_hip_z | hinge | angular velocity ($rad/s$) |
| 31 | Angular velocity of the left hip (y-coordinate) | -Inf | Inf | jointvel_left_hip_y | hinge | angular velocity ($rad/s$) |
| 32 | Angular velocity of the left leg | -Inf | Inf | jointvel_left_leg | hinge | angular velocity ($rad/s$) |
| 33 | Angular velocity of the left foot | -Inf | Inf | jointvel_left_foot | hinge | angular velocity ($rad/s$) |

Table 7: Action space of Humanoid

| Num | Action | Min | Max | Name in XML | Joint | Unit |
|---|---|---|---|---|---|---|
| 0 | Torque applied on the rotor between torso and the right hip (x-coordinate) | -1 | 1 | right_hip_x | hinge | torque ($Nm$) |
| 1 | Torque applied on the rotor between torso and the right hip (z-coordinate) | -1 | 1 | right_hip_z | hinge | torque ($Nm$) |
| 2 | Torque applied on the rotor between torso and the right hip (y-coordinate) | -1 | 1 | right_hip_y | hinge | torque ($Nm$) |
| 3 | Torque applied on the right knee rotor | -1 | 1 | right_knee | hinge | torque ($Nm$) |
| 4 | Torque applied on the rotor between torso and the left hip (x-coordinate) | -1 | 1 | left_hip_x | hinge | torque ($Nm$) |
| 5 | Torque applied on the rotor between torso and the left hip (z-coordinate) | -1 | 1 | left_hip_z | hinge | torque ($Nm$) |
| 6 | Torque applied on the rotor between torso and the left hip (y-coordinate) | -1 | 1 | left_hip_y | hinge | torque ($Nm$) |
| 7 | Torque applied on the left knee rotor | -1 | 1 | left_knee | hinge | torque ($Nm$) |

Table 8: Observation space of Humanoid

| Num | Observation | Min | Max | Name in XML | Joint | Unit |
|---|---|---|---|---|---|---|
| 0/1/2 | 3-axis linear acceleration of the torso (including gravity) | -Inf | Inf | accelerometer | free | acceleration ($m/s^2$) |
| 3/4/5 | 3-axis linear velocity of the torso | -Inf | Inf | velocimeter | free | velocity ($m/s$) |
| 6/7/8 | 3-axis angular velocity of the torso | -Inf | Inf | gyro | free | angular velocity ($rad/s$) |
| 9/10/11 | 3D magnetic flux vector at the torso | -Inf | Inf | magnetometer | free | magnetic flux density ($T$) |
| 12 | Contact force at the right foot | 0 | Inf | touch_right_foot | - | force($N$) |
| 13 | Contact force at the left foot | 0 | Inf | touch_left_foot | - | force($N$) |
| 14 | Angle of the right hip (x-coordinate) | -Inf | Inf | jointpos_right_hip_x | hinge | angle ($rad$) |
| 15 | Angle of the right hip (z-coordinate) | -Inf | Inf | jointpos_right_hip_z | hinge | angle ($rad$) |
| 16 | Angle of the right hip (y-coordinate) | -Inf | Inf | jointpos_right_hip_y | hinge | angle ($rad$) |
| 17 | Angle of the right knee | -Inf | Inf | jointpos_right_knee | hinge | angle ($rad$) |
| 18 | Angle of the left hip (x-coordinate) | -Inf | Inf | jointpos_left_hip_x | hinge | angle ($rad$) |
| 19 | Angle of the left hip (z-coordinate) | -Inf | Inf | jointpos_left_hip_z | hinge | angle ($rad$) |
| 20 | Angle of the left hip (y-coordinate) | -Inf | Inf | jointpos_left_hip_y | hinge | angle ($rad$) |
| 21 | Angle of the left leg | -Inf | Inf | jointpos_left_knee | hinge | angle ($rad$) |
| 22 | Angular velocity of the right hip (x-coordinate) | -Inf | Inf | jointvel_right_hip_x | hinge | angular velocity ($rad/s$) |
| 23 | Angular velocity of the right hip (z-coordinate) | -Inf | Inf | jointvel_right_hip_z | hinge | angular velocity ($rad/s$) |
| 24 | Angular velocity of the right hip (y-coordinate) | -Inf | Inf | jointvel_right_hip_y | hinge | angular velocity ($rad/s$) |
| 25 | Angular velocity of the right knee | -Inf | Inf | jointvel_right_knee | hinge | angular velocity ($rad/s$) |
| 26 | Angular velocity of the left hip (x-coordinate) | -Inf | Inf | jointvel_left_hip_x | hinge | angular velocity ($rad/s$) |
| 27 | Angular velocity of the left hip (z-coordinate) | -Inf | Inf | jointvel_left_hip_z | hinge | angular velocity ($rad/s$) |
| 28 | Angular velocity of the left hip (y-coordinate) | -Inf | Inf | jointvel_left_hip_y | hinge | angular velocity ($rad/s$) |
| 29 | Angular velocity of the left knee | -Inf | Inf | jointvel_left_knee | hinge | angular velocity ($rad/s$) |

Table 9: Action space of Hopper

| Num | Action | Min | Max | Name in XML | Joint | Unit |
|---|---|---|---|---|---|---|
| 0 | Torque applied on the rotor between torso and the hip (x-coordinate) | -1 | 1 | hip_x | hinge | torque ($Nm$) |
| 1 | Torque applied on the rotor between torso and the hip (z-coordinate) | -1 | 1 | hip_z | hinge | torque ($Nm$) |
| 2 | Torque applied on the rotor between torso and the hip (y-coordinate) | -1 | 1 | hip_y | hinge | torque ($Nm$) |
| 3 | Torque applied on the thigh rotor | -1 | 1 | thigh_joint | hinge | torque ($Nm$) |
| 4 | Torque applied on the leg rotor | -1 | 1 | leg_joint | hinge | torque ($Nm$) |
| 5 | Torque applied on the foot rotor | -1 | 1 | foot_joint | hinge | torque ($Nm$) |

Table 10: Observation space of Hopper

| Num | Observation | Min | Max | Name in XML | Joint | Unit |
|---|---|---|---|---|---|---|
| 0/1/2 | 3-axis linear acceleration of the torso (including gravity) | -Inf | Inf | accelerometer | free | acceleration ($m/s^2$) |
| 3/4/5 | 3-axis linear velocity of the torso | -Inf | Inf | velocimeter | free | velocity ($m/s$) |
| 6/7/8 | 3-axis angular velocity of the torso | -Inf | Inf | gyro | free | angular velocity ($rad/s$) |
| 9/10/11 | 3D magnetic flux vector at the torso | -Inf | Inf | magnetometer | free | magnetic flux density ($T$) |
| 12 | Contact force at the foot | 0 | Inf | touch_foot | - | force($N$) |
| 13 | Angle of the hip (x-coordinate) | -Inf | Inf | jointpos_hip_x | hinge | angle ($rad$) |
| 14 | Angle of the hip (z-coordinate) | -Inf | Inf | jointpos_hip_z | hinge | angle ($rad$) |
| 15 | Angle of the hip (y-coordinate) | -Inf | Inf | jointpos_hip_y | hinge | angle ($rad$) |
| 16 | Angle of the thigh | -Inf | Inf | jointpos_thigh | hinge | angle ($rad$) |
| 17 | Angle of the leg | -Inf | Inf | jointpos_leg | hinge | angle ($rad$) |
| 18 | Angle of the foot | -Inf | Inf | jointpos_foot | hinge | angle ($rad$) |
| 19 | Angular velocity of the hip (x-coordinate) | -Inf | Inf | jointvel_hip_x | hinge | angular velocity ($rad/s$) |
| 20 | Angular velocity of the hip (z-coordinate) | -Inf | Inf | jointvel_hip_z | hinge | angular velocity ($rad/s$) |
| 21 | Angular velocity of the hip (y-coordinate) | -Inf | Inf | jointvel_hip_y | hinge | angular velocity ($rad/s$) |
| 22 | Angular velocity of the thigh | -Inf | Inf | jointvel_thigh | hinge | angular velocity ($rad/s$) |
| 23 | Angular velocity of the leg | -Inf | Inf | jointvel_leg | hinge | angular velocity ($rad/s$) |
| 24 | Angular velocity of the foot | -Inf | Inf | jointvel_foot | hinge | angular velocity ($rad/s$) |

Table 11: Action space of Arm3

| Num | Action | Min | Max | Name in XML | Joint | Unit |
|---|---|---|---|---|---|---|
| 0 | Torque applied on the first joint (connecting the base point and the first link) | -1 | 1 | joint_1 | hinge | torque ($Nm$) |
| 1 | Torque applied on the second joint(connecting the first and the second link) | -1 | 1 | joint_2 | hinge | torque ($Nm$) |
| 2 | Torque applied on the third joint (connecting the second and the third link) | -1 | 1 | joint_3 | hinge | torque ($Nm$) |

Table 12: Observation space of Arm3

| Num | Observation | Min | Max | Name in XML | Joint | Unit |
|---|---|---|---|---|---|---|
| 0/1/2 | 3-axis linear acceleration of the first link (including gravity) | -Inf | Inf | accelerometer_link_1 | free | acceleration ($m/s^2$) |
| 3/4/5 | 3-axis linear velocity of the first link | -Inf | Inf | velocimeter_link_1 | free | velocity ($m/s$) |
| 6/7/8 | 3-axis angular velocity velocity of the first link | -Inf | Inf | gyro_link_1 | free | angular velocity ($rad/s$) |
| 9/10/11 | 3D magnetic flux vector at the first link | -Inf | Inf | magnetometer_link_1 | free | magnetic flux density ($T$) |
| 12/13/14 | 3-axis linear acceleration of the second link (including gravity) | -Inf | Inf | accelerometer_link_2 | free | acceleration ($m/s^2$) |
| 15/16/17 | 3-axis linear velocity of the second link | -Inf | Inf | velocimeter_link_2 | free | velocity ($m/s$) |
| 18/19/20 | 3-axis angular velocity velocity of the second link | -Inf | Inf | gyro_link_2 | free | angular velocity ($rad/s$) |
| 21/22/23 | 3D magnetic flux vector at the second link | -Inf | Inf | magnetometer_link_2 | free | magnetic flux density ($T$) |
| 24/25/26 | 3-axis linear acceleration of the third link (including gravity) | -Inf | Inf | accelerometer_link_3 | free | acceleration ($m/s^2$) |
| 27/28/29 | 3-axis linear velocity of the third link | -Inf | Inf | velocimeter_link_3 | free | velocity ($m/s$) |
| 30/31/32 | 3-axis angular velocity velocity of the third link | -Inf | Inf | gyro_link_3 | free | angular velocity ($rad/s$) |
| 33/34/35 | 3D magnetic flux vector at the third link | -Inf | Inf | magnetometer_link_3 | free | magnetic flux density ($T$) |
| 36/37/38 | 3-axis linear acceleration of the fourth link (including gravity) | -Inf | Inf | accelerometer_link_4 | free | acceleration ($m/s^2$) |
| 39/40/41 | 3-axis linear velocity of the fourth link | -Inf | Inf | velocimeter_link_4 | free | velocity ($m/s$) |
| 42/43/44 | 3-axis angular velocity velocity of the fourth link | -Inf | Inf | gyro_link_4 | free | angular velocity ($rad/s$) |
| 45/46/47 | 3D magnetic flux vector at the fourth link | -Inf | Inf | magnetometer_link_4 | free | magnetic flux density ($T$) |
| 48/49/50 | 3-axis linear acceleration of the fifth link (including gravity) | -Inf | Inf | accelerometer_link_5 | free | acceleration ($m/s^2$) |
| 51/52/53 | 3-axis linear velocity of the fifth link | -Inf | Inf | velocimeter_link_5 | free | velocity ($m/s$) |
| 54/55/56 | 3-axis angular velocity velocity of the fifth link | -Inf | Inf | gyro_link_5 | free | angular velocity ($rad/s$) |
| 57/58/59 | 3D magnetic flux vector at the fifth link | -Inf | Inf | magnetometer_link_5 | free | magnetic flux density ($T$) |
| 60 | Angle of the first joint | -Inf | Inf | jointpos_joint_1 | hinge | angle ($rad$) |
| 61 | Angle of the second joint | -Inf | Inf | jointpos_joint_2 | hinge | angle ($rad$) |
| 62 | Angle of the third joint | -Inf | Inf | jointpos_joint_3 | hinge | angle ($rad$) |
| 63 | Angular velocity of the first joint | -Inf | Inf | jointvel_joint_1 | hinge | angular velocity ($rad/s$) |
| 64 | Angular velocity of the second joint | -Inf | Inf | jointvel_joint_2 | hinge | angular velocity ($rad/s$) |
| 65 | Angular velocity of the third joint | -Inf | Inf | jointvel_joint_3 | hinge | angular velocity ($rad/s$) |
| 66 | Contact force at the end effector | 0 | Inf | touch_end_effector | - | force($N$) |

Table 13: Action space of Arm6

| Num | Action | Min | Max | Name in XML | Joint | Unit |
|---|---|---|---|---|---|---|
| 0 | Torque applied on the first joint (connecting the base point and the first link) | -1 | 1 | joint_1 | hinge | torque ($Nm$) |
| 1 | Torque applied on the second joint(connecting the first and the second link) | -1 | 1 | joint_2 | hinge | torque ($Nm$) |
| 2 | Torque applied on the third joint (connecting the second and the third link) | -1 | 1 | joint_3 | hinge | torque ($Nm$) |
| 3 | Torque applied on the fourth joint (connecting the third and the fourth link) | -1 | 1 | joint_4 | hinge | torque ($Nm$) |
| 4 | Torque applied on the fifth joint (connecting the fourth and the fifth link) | -1 | 1 | joint_5 | hinge | torque ($Nm$) |
| 5 | Torque applied on the sixth joint (connecting the fifth and the sixth link) | -1 | 1 | joint_6 | hinge | torque ($Nm$) |

Table 14: Observation space of Arm6

| Num | Observation | Min | Max | Name in XML | Joint | Unit |
|---|---|---|---|---|---|---|
| 0/1/2 | 3-axis linear acceleration of the first link (including gravity) | -Inf | Inf | accelerometer_link_1 | free | acceleration ($m/s^2$) |
| 3/4/5 | 3-axis linear velocity of the first link | -Inf | Inf | velocimeter_link_1 | free | velocity ($m/s$) |
| 6/7/8 | 3-axis angular velocity of the first link | -Inf | Inf | gyro_link_1 | free | angular velocity ($rad/s$) |
| 9/10/11 | 3D magnetic flux vector at the first link | -Inf | Inf | magnetometer_link_1 | free | magnetic flux density ($T$) |
| 12/13/14 | 3-axis linear acceleration of the second link (including gravity) | -Inf | Inf | accelerometer_link_2 | free | acceleration ($m/s^2$) |
| 15/16/17 | 3-axis linear velocity of the second link | -Inf | Inf | velocimeter_link_2 | free | velocity ($m/s$) |
| 18/19/20 | 3-axis angular velocity velocity of the second link | -Inf | Inf | gyro_link_2 | free | angular velocity ($rad/s$) |
| 21/22/23 | 3D magnetic flux vector at the second link | -Inf | Inf | magnetometer_link_2 | free | magnetic flux density ($T$) |
| 24/25/26 | 3-axis linear acceleration of the third link (including gravity) | -Inf | Inf | accelerometer_link_3 | free | acceleration ($m/s^2$) |
| 27/28/29 | 3-axis linear velocity of the third link | -Inf | Inf | velocimeter_link_3 | free | velocity ($m/s$) |
| 30/31/32 | 3-axis angular velocity velocity of the third link | -Inf | Inf | gyro_link_3 | free | angular velocity ($rad/s$) |
| 33/34/35 | 3D magnetic flux vector at the third link | -Inf | Inf | magnetometer_link_3 | free | magnetic flux density ($T$) |
| 36/37/38 | 3-axis linear acceleration of the fourth link (including gravity) | -Inf | Inf | accelerometer_link_4 | free | acceleration ($m/s^2$) |
| 39/40/41 | 3-axis linear velocity of the fourth link | -Inf | Inf | velocimeter_link_4 | free | velocity ($m/s$) |
| 42/43/44 | 3-axis angular velocity velocity of the fourth link | -Inf | Inf | gyro_link_4 | free | angular velocity ($rad/s$) |
| 45/46/47 | 3D magnetic flux vector at the fourth link | -Inf | Inf | magnetometer_link_4 | free | magnetic flux density ($T$) |
| 48/49/50 | 3-axis linear acceleration of the fifth link (including gravity) | -Inf | Inf | accelerometer_link_5 | free | acceleration ($m/s^2$) |
| 51/52/53 | 3-axis linear velocity of the fifth link | -Inf | Inf | velocimeter_link_5 | free | velocity ($m/s$) |
| 54/55/56 | 3-axis angular velocity velocity of the fifth link | -Inf | Inf | gyro_link_5 | free | angular velocity ($rad/s$) |
| 57/58/59 | 3D magnetic flux vector at the fifth link | -Inf | Inf | magnetometer_link_5 | free | magnetic flux density ($T$) |
| 60/61/62 | 3-axis linear acceleration of the sixth link (including gravity) | -Inf | Inf | accelerometer_link_6 | free | acceleration ($m/s^2$) |
| 63/64/65 | 3-axis linear velocity of the sixth link | -Inf | Inf | velocimeter_link_6 | free | velocity ($m/s$) |
| 66/67/68 | 3-axis angular velocity velocity of the sixth link | -Inf | Inf | gyro_link_6 | free | angular velocity ($rad/s$) |
| 69/70/71 | 3D magnetic flux vector at the sixth link | -Inf | Inf | magnetometer_link_6 | free | magnetic flux density ($T$) |
| 72/73/74 | 3-axis linear acceleration of the seventh link (including gravity) | -Inf | Inf | accelerometer_link_7 | free | acceleration ($m/s^2$) |
| 75/76/77 | 3-axis linear velocity of the seventh link | -Inf | Inf | velocimeter_link_7 | free | velocity ($m/s$) |
| 78/79/80 | 3-axis angular velocity velocity of the seventh link | -Inf | Inf | gyro_link_7 | free | angular velocity ($rad/s$) |
| 81/82/83 | 3D magnetic flux vector at the seventh link | -Inf | Inf | magnetometer_link_7 | free | magnetic flux density ($T$) |
| 84 | Angle of the first joint | -Inf | Inf | jointpos_joint_1 | hinge | angle ($rad$) |
| 85 | Angle of the second joint | -Inf | Inf | jointpos_joint_2 | hinge | angle ($rad$) |
| 86 | Angle of the third joint | -Inf | Inf | jointpos_joint_3 | hinge | angle ($rad$) |
| 87 | Angle of the fourth joint | -Inf | Inf | jointpos_joint_4 | hinge | angle ($rad$) |
| 88 | Angle of the fifth joint | -Inf | Inf | jointpos_joint_5 | hinge | angle ($rad$) |
| 89 | Angle of the sixth joint | -Inf | Inf | jointpos_joint_6 | hinge | angle ($rad$) |
| 90 | Angular velocity of the first joint | -Inf | Inf | jointvel_joint_1 | hinge | angular velocity ($rad/s$) |
| 91 | Angular velocity of the second joint | -Inf | Inf | jointvel_joint_2 | hinge | angular velocity ($rad/s$) |
| 92 | Angular velocity of the third joint | -Inf | Inf | jointvel_joint_3 | hinge | angular velocity ($rad/s$) |
| 93 | Angular velocity of the fourth joint | -Inf | Inf | jointvel_joint_4 | hinge | angular velocity ($rad/s$) |
| 94 | Angular velocity of the fifth joint | -Inf | Inf | jointvel_joint_5 | hinge | angular velocity ($rad/s$) |
| 95 | Angular velocity of the sixth joint | -Inf | Inf | jointvel_joint_6 | hinge | angular velocity ($rad/s$) |
| 96 | Contact force at the end effector | 0 | Inf | touch_end_effector | - | force($N$) |

Table 15: Action space of Drone

| Num | Action | Min | Max | Name in XML | Joint | Unit |
|---|---|---|---|---|---|---|
| 0 | Extra thrust force applied on the first propeller | -1 | 1 | - | - | force $(N)$ |
| 1 | Extra thrust force applied on the second propeller | -1 | 1 | - | - | force $(N)$ |
| 2 | Extra thrust force applied on the third propeller | -1 | 1 | - | - | force $(N)$ |
| 3 | Extra thrust force applied on the fourth propeller | -1 | 1 | - | - | force $(N)$ |

Table 16: Observation space of Drone

| Num | Observation | Min | Max | Name in XML | Joint | Unit |
|---|---|---|---|---|---|---|
| 0/1/2 | 3-axis linear acceleration of the torso (including gravity) | -Inf | Inf | accelerometer | free | acceleration $(m/s^2)$ |
| 3/4/5 | 3-axis linear velocity of the torso | -Inf | Inf | velocimeter | free | velocity $(m/s)$ |
| 6/7/8 | 3-axis angular velocity velocity of the torso | -Inf | Inf | gyro | free | angular velocity $(rad/s)$ |
| 9/10/11 | 3D magnetic flux vector at the torso | -Inf | Inf | magnetometer | free | magnetic flux density $(T)$ |
| 12 | Contact force at the upper point of the first propeller | 0 | Inf | touch_p1a | - | force$(N)$ |
| 13 | Contact force at the lower point of the first propeller | 0 | Inf | touch_p1b | - | force$(N)$ |
| 14 | Contact force at the upper point of the second propeller | 0 | Inf | touch_p2a | - | force$(N)$ |
| 15 | Contact force at the lower point of the second propeller | 0 | Inf | touch_p2b | - | force$(N)$ |
| 16 | Contact force at the upper point of the third propeller | 0 | Inf | touch_p3a | - | force$(N)$ |
| 17 | Contact force at the lower point of the third propeller | 0 | Inf | touch_p3b | - | force$(N)$ |
| 18 | Contact force at the upper point of the fourth propeller | 0 | Inf | touch_p4a | - | force$(N)$ |
| 19 | Contact force at the lower point of the fourth propeller | 0 | Inf | touch_p4b | - | force$(N)$ |

## B.2 Observation Space Options and Desiderata

The observation spaces are also updated to match the new 3D tasks. The 3D compasses and 3D pseudo-lidars are introduced for 3D robots to sensor the position of targets in 3D space. Different from the single lidar system of the original environment, the Advanced Safety Gym allows multiple lidars on different parts of the robot. For example, in Figure 13a the Arm robot is equipped with a 3D lidar and a 3D compass on each joint to obtain more environment information. Figure 13b shows a drone equipped with two 3D lidars to observe the 3D hazards and the 3D goal. The "lidar halos" of two lidars are distributed on two spheres with different radii. The number of "lidar halos" is configurable for more dense observations.

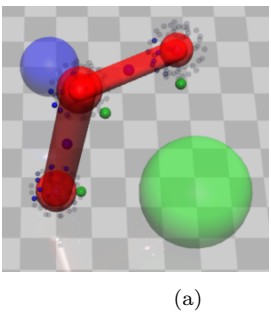
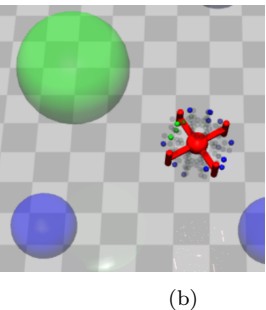

(a)  (b)

Figure 13: Visualizations of observation spaces

## B.3 Layout Randomization Options and Desiderata

The layout randomization is inherited from the original Safety Gym. In order to generate 3D objects, the $z$ coordinate can be configured or randomly picked after the $x$ and $y$ coordinates are generated.

## B.4 Task and Constraint Details

Table 17: Comparison between different tasks

|  | GUARD Tasks | | | | SafetyGym Tasks | | |
|  | Goal | Push | Chase | Defense | Goal | Button | Push |
| --- | --- | --- | --- | --- | --- | --- | --- |
| Interactive task |  | ✓ | ✓ | ✓ |  |  | ✓ |
| Non-interactive task | ✓ |  |  |  | ✓ | ✓ |  |
| Contact task |  | ✓ | ✓ | ✓ |  | ✓ | ✓ |
| Non-contact task | ✓ |  | ✓ | ✓ | ✓ |  |  |
| 2D task | ✓ | ✓ | ✓ | ✓ | ✓ | ✓ | ✓ |
| 3D task | ✓ |  | ✓ | ✓ |  |  |  |
| Movable target |  | ✓ | ✓ | ✓ |  |  | ✓ |
| Immovable target | ✓ |  |  |  | ✓ | ✓ |  |
| Single target | ✓ | ✓ | ✓ | ✓ | ✓ | ✓ | ✓ |
| Multiple targets |  |  | ✓ | ✓ |  |  |  |
| General contact target |  | ✓ | ✓ | ✓ |  | ✓ |  |

Table 18: Comparison between different constraints

| | New Constraints | | | Inherited Constraints | | | | |
|---|---|---|---|---|---|---|---|---|
| | Ghosts | Ghosts 3D | Hazards 3D | Hazards | Vases | Pillars | Buttons | Gremlins |
| Trespassable | ✓ | ✓ | | | ✓ | ✓ | ✓ | ✓ |
| Untrespassable | ✓ | ✓ | ✓ | ✓ | | | | |
| Immovable | | | ✓ | ✓ | | ✓ | ✓ | |
| Passively movable | | | | | ✓ | | | |
| Actively movable | ✓ | ✓ | | | | | | ✓ |
| 3D motion | | ✓ | ✓ | | | | | |
| 2D motion | ✓ | ✓ | ✓ | ✓ | ✓ | ✓ | ✓ | ✓ |

## B.5 Dynamics of movable objects

We begin by defining the distance vector $d_{\mathrm{origin}} = x_{\mathrm{origin}} - x_{\mathrm{object}}$, which represents the distance from the position of the dynamic object $x_{\mathrm{object}}$ to the origin point of the world framework $x_{\mathrm{origin}}$. By default, the origin point is set to $(0, 0, 0)$. Next, we define the distance vector $d_{\mathrm{robot}} = x_{\mathrm{robot}} - x_{\mathrm{object}}$, which represents the distance from the dynamic object $x_{\mathrm{object}}$ to the position of the robot $x_{\mathrm{robot}}$. We introduce two parameters: $r_0$, which defines a circular area centered at the origin point within which the objects are limited to move. $r_1$, which represents the threshold distance that the dynamic objects strive to maintain from the robot. Finally, we have three configurable non-negative velocity constants for the dynamic objects: $v_0$, $v_1$, and $v_2$.

### B.5.1 Dynamics of targets of Chase task

$$\dot{x}_{object} = \begin{cases} v_0*d_{origin}, & \text{if } \|d_{origin}\| > r_0 \\ -v_1*d_{robot}, & \text{if } \|d_{origin}\| \le r_0 \text{ and } \|d_{robot}\| \le r_1 \\ 0, & \text{if } \|d_{origin}\| \le r_0 \text{ and } \|d_{robot}\| > r_1 \end{cases}, \tag{11}$$

### B.5.2 Dynamics of targets of Defense task

$$\dot{x}_{object} = \begin{cases} v_0*d_{origin}, & \text{if } \|d_{origin}\| > r_0 \\ -v_1*d_{robot}, & \text{if } \|d_{origin}\| \le r_0 \text{ and } \|d_{robot}\| \le r_1 \\ v_2*d_{origin}, & \text{if } \|d_{origin}\| \le r_0 \text{ and } \|d_{robot}\| > r_1 \end{cases}, \tag{12}$$

### B.5.3 Dynamics of ghost and 3D ghost

$$\dot{x}_{object} = \begin{cases} v_0*d_{origin}, & \text{if } \|d_{origin}\| > r_0 \\ v_1*d_{robot}, & \text{if } \|d_{origin}\| \le r_0 \text{ and } \|d_{robot}\| > r_1 \\ 0, & \text{if } \|d_{origin}\| \le r_0 \text{ and } \|d_{robot}\| \le r_1 \end{cases}, \tag{13}$$

## C    Experiment Details

The GUARD implementation is partially inspired by Safety Gym (Ray et al., 2019b) and Spinningup (Achiam, 2018) which are both under MIT license.

### C.1    Policy Settings

The hyper-parameters used in our experiments are listed in Table 19 as default.

Our experiments use separate multilayer perceptrons with *tanh* activations for the policy network, value network and cost network. Each network consists of two hidden layers of size (64,64). All of the networks are trained using *Adam* optimizer with a learning rate of 0.01.

We apply an on-policy framework in our experiments. During each epoch the agent interacts $B$ times with the environment and then performs a policy update based on the experience collected from the current epoch. The maximum length of the trajectory is set to 1000 and the total epoch number $N$ is set to 200 as default. In our experiments the Walker and the Ant were trained for 1000 epochs due to the high dimension.

The policy update step is based on the scheme of TRPO, which performs up to 100 steps of backtracking with a coefficient of 0.8 for line searching.

For all experiments, we use a discount factor of $\gamma = 0.99$, an advantage discount factor $\lambda = 0.95$, and a KL-divergence step size of $\delta_{KL} = 0.02$.

For experiments which consider cost constraints we adopt a target cost $\delta_c = 0.0$ to pursue a zero-violation policy.

Other unique hyper-parameters for each algorithm are hand-tuned to attain reasonable performance.

Each model is trained on a server with a 48-core Intel(R) Xeon(R) Silver 4214 CPU @ 2.2.GHz, Nvidia RTX A4000 GPU with 16GB memory, and Ubuntu 20.04.

For low-dimensional tasks, we train each model for 6e6 steps which takes around seven hours. For high-dimensional tasks, we train each model for 3e7 steps which takes around 60 hours.

### C.2    Experiment tasks

The experiment settings, detailed in Table 20, encompass a total of 72 combinations derived from 4 task types, 9 robot types, and 2 constraint types. For the purpose of comprehensive comparison in this paper, we have selected 36 experiments involving 8 hazards and 9 experiments featuring 8 ghosts.

### C.3    Metrics Comparison

we report all the 45 results of our test suites by three metrics:

- The average episode returns $J_r$.

- The average episodic sum of costs $M_c$.

- The average cost over the entirety of training $\rho_c$.

All of the three metrics were obtained from the final epoch after convergence. Each metric was averaged over two random seeds.

Table 19: Important hyper-parameters of different algorithms in our experiments

| Policy Parameter | | TRPO | TRPO-Lagrangian | TRPO-SL [18' Dalal] | TRPO-USL | TRPO-IPO | TRPO-FAC | CPO | PCPO |
|---|---|---|---|---|---|---|---|---|---|
| Epochs | $N$ | 200 | 200 | 200 | 200 | 200 | 200 | 200 | 200 |
| Steps per epoch | $B$ | 30000 | 30000 | 30000 | 30000 | 30000 | 30000 | 30000 | 30000 |
| Maximum length of trajectory | $L$ | 1000 | 1000 | 1000 | 1000 | 1000 | 1000 | 1000 | 1000 |
| Policy network hidden layers | | (64, 64) | (64, 64) | (64, 64) | (64, 64) | (64, 64) | (64, 64) | (64, 64) | (64, 64) |
| Discount factor | $\gamma$ | 0.99 | 0.99 | 0.99 | 0.99 | 0.99 | 0.99 | 0.99 | 0.99 |
| Advantage discount factor | $\lambda$ | 0.97 | 0.97 | 0.97 | 0.97 | 0.97 | 0.97 | 0.97 | 0.97 |
| TRPO backtracking steps | | 100 | 100 | 100 | 100 | 100 | 100 | 100 | - |
| TRPO backtracking coefficient | | 0.8 | 0.8 | 0.8 | 0.8 | 0.8 | 0.8 | 0.8 | - |
| Target KL | $\delta_{KL}$ | 0.02 | 0.02 | 0.02 | 0.02 | 0.02 | 0.02 | 0.02 | 0.02 |
| Value network hidden layers | | (64, 64) | (64, 64) | (64, 64) | (64, 64) | (64, 64) | (64, 64) | (64, 64) | (64, 64) |
| Value network iteration | | 80 | 80 | 80 | 80 | 80 | 80 | 80 | 80 |
| Value network optimizer | | Adam | Adam | Adam | Adam | Adam | Adam | Adam | Adam |
| Value learning rate | | 0.001 | 0.001 | 0.001 | 0.001 | 0.001 | 0.001 | 0.001 | 0.001 |
| Cost network hidden layers | | - | (64, 64) | (64, 64) | (64, 64) | - | (64, 64) | (64, 64) | (64, 64) |
| Cost network iteration | | - | 80 | 80 | 80 | - | 80 | 80 | 80 |
| Cost network optimizer | | - | Adam | Adam | Adam | - | Adam | Adam | Adam |
| Cost learning rate | | - | 0.001 | 0.001 | 0.001 | - | 0.001 | 0.001 | 0.001 |
| Target Cost | $\delta_c$ | - | 0.0 | 0.0 | 0.0 | 0.0 | 0.0 | 0.0 | 0.0 |
| Lagrangian optimizer | | - | - | - | - | - | Adam | - | - |
| Lagrangian learning rate | | - | 0.005 | - | - | - | 0.0001 | - | - |
| USL correction iteration | | - | - | - | 20 | - | - | - | - |
| USL correction rate | | - | - | - | 0.05 | - | - | - | - |
| Warmup ratio | | - | - | 1/3 | 1/3 | - | - | - | - |
| IPO parameter | $t$ | - | - | - | - | 0.01 | - | - | - |
| Cost reduction | | - | - | - | - | - | - | 0.0 | - |

| Goal_Point_8Hazards |
| :---: |
| Goal_Point_8Ghosts |
| Goal_Swimmer_8Hazards |
| Goal_Swimmer_8Ghosts |
| Goal_Ant_8Hazards |
| Goal_Ant_8Ghosts |
| Goal_Walker_8Hazards |
| Goal_Walker_8Ghosts |
| Goal_Humanoid_8Hazards |
| Goal_Humanoid_8Ghosts |
| Goal_Hopper_8Hazards |
| Goal_Hopper_8Ghosts |
| Goal_Arm3_8Hazards |
| Goal_Arm3_8Ghosts |
| Goal_Arm6_8Hazards |
| Goal_Arm6_8Ghosts |
| Goal_Drone_8Hazards |
| Goal_Drone_8Ghosts |

(a) Goal

| Push_Point_8Hazards |
| :---: |
| Push_Point_8Ghosts |
| Push_Swimmer_8Hazards |
| Push_Swimmer_8Ghosts |
| Push_Ant_8Hazards |
| Push_Ant_8Ghosts |
| Push_Walker_8Hazards |
| Push_Walker_8Ghosts |
| Push_Humanoid_8Hazards |
| Push_Humanoid_8Ghosts |
| Push_Hopper_8Hazards |
| Push_Hopper_8Ghosts |
| Push_Arm3_8Hazards |
| Push_Arm3_8Ghosts |
| Push_Arm6_8Hazards |
| Push_Arm6_8Ghosts |
| Push_Drone_8Hazards |
| Push_Drone_8Ghosts |

(b) Push

| Chase_Point_8Hazards |
| :---: |
| Chase_Point_8Ghosts |
| Chase_Swimmer_8Hazards |
| Chase_Swimmer_8Ghosts |
| Chase_Ant_8Hazards |
| Chase_Ant_8Ghosts |
| Chase_Walker_8Hazards |
| Chase_Walker_8Ghosts |
| Chase_Humanoid_8Hazards |
| Chase_Humanoid_8Ghosts |
| Chase_Hopper_8Hazards |
| Chase_Hopper_8Ghosts |
| Chase_Arm3_8Hazards |
| Chase_Arm3_8Ghosts |
| Chase_Arm6_8Hazards |
| Chase_Arm6_8Ghosts |
| Chase_Drone_8Hazards |
| Chase_Drone_8Ghosts |

(c) Chase

| Defense_Point_8Hazards |
| :---: |
| Defense_Point_8Ghosts |
| Defense_Swimmer_8Hazards |
| Defense_Swimmer_8Ghosts |
| Defense_Ant_8Hazards |
| Defense_Ant_8Ghosts |
| Defense_Walker_8Hazards |
| Defense_Walker_8Ghosts |
| Defense_Humanoid_8Hazards |
| Defense_Humanoid_8Ghosts |
| Defense_Hopper_8Hazards |
| Defense_Hopper_8Ghosts |
| Defense_Arm3_8Hazards |
| Defense_Arm3_8Ghosts |
| Defense_Arm6_8Hazards |
| Defense_Arm6_8Ghosts |
| Defense_Drone_8Hazards |
| Defense_Drone_8Ghosts |

(d) Defense

Table 20: Tasks of our environments

Table 21: Metrics of nine **Goal_{Robot}_8Hazards** environments obtained from the final epoch.

Goal_Point_8Hazards

| Algorithm | $\bar{J}_r$ | $\bar{M}_c$ | $\bar{\rho}_c$ |
|---|---|---|---|
| TRPO | 26.2296 | 7.4550 | 0.0067 |
| TRPO-Lagrangian | 25.4503 | 2.5031 | **0.0034** |
| TRPO-SL | 19.0765 | 3.5200 | 0.0056 |
| TRPO-USL | 24.6524 | 7.0004 | 0.0060 |
| TRPO-IPO | 20.3057 | 4.4037 | 0.0049 |
| TRPO-FAC | **26.9707** | **2.1581** | 0.0038 |
| CPO | 25.9157 | 3.2388 | 0.0036 |
| PCPO | 24.9032 | 3.7118 | 0.0048 |

Goal_Swimmer_8Hazards

| Algorithm | $\bar{J}_r$ | $\bar{M}_c$ | $\bar{\rho}_c$ |
|---|---|---|---|
| TRPO | **31.5282** | 11.4067 | 0.0117 |
| TRPO-Lagrangian | 19.5685 | **4.3231** | **0.0074** |
| TRPO-SL | 9.2362 | 4.4453 | 0.0075 |
| TRPO-USL | 30.2756 | 10.2352 | 0.0100 |
| TRPO-IPO | 9.5714 | 7.9993 | 0.0079 |
| TRPO-FAC | 24.8486 | 7.8014 | 0.0085 |
| CPO | 26.6166 | 9.2452 | 0.0095 |
| PCPO | 24.4054 | 9.3452 | 0.0094 |

Goal_Ant_8Hazards

| Algorithm | $\bar{J}_r$ | $\bar{M}_c$ | $\bar{\rho}_c$ |
|---|---|---|---|
| TRPO | 59.3694 | 7.9737 | 0.0097 |
| TRPO-Lagrangian | 35.0180 | **2.7954** | **0.0056** |
| TRPO-SL | 24.0752 | 45.9755 | 0.0355 |
| TRPO-USL | 59.2213 | 9.2237 | 0.0096 |
| TRPO-IPO | 2.6040 | 6.3006 | 0.0059 |
| TRPO-FAC | 48.2685 | 5.6736 | 0.0071 |
| CPO | 60.2093 | 8.1194 | 0.0092 |
| PCPO | **60.3654** | 8.9137 | 0.0091 |

Goal_Walker_8Hazards

| Algorithm | $\bar{J}_r$ | $\bar{M}_c$ | $\bar{\rho}_c$ |
|---|---|---|---|
| TRPO | 56.7139 | 9.8112 | 0.0104 |
| TRPO-Lagrangian | 33.7839 | **3.3714** | **0.0053** |
| TRPO-SL | 39.9848 | 12.7370 | 0.0128 |
| TRPO-USL | **57.1097** | 9.9469 | 0.0097 |
| TRPO-IPO | 7.2728 | 6.7115 | 0.0068 |
| TRPO-FAC | 42.6250 | 4.4426 | 0.0062 |
| CPO | 51.9246 | 8.0409 | 0.0082 |
| PCPO | 55.0100 | 10.0377 | 0.0089 |

Goal_Humanoid_8Hazards

| Algorithm | $\bar{J}_r$ | $\bar{M}_c$ | $\bar{\rho}_c$ |
|---|---|---|---|
| TRPO | 11.6758 | 8.2332 | 0.0079 |
| TRPO-Lagrangian | 6.1294 | 7.6847 | **0.0066** |
| TRPO-SL | 9.1517 | 10.1473 | 0.0091 |
| TRPO-USL | 10.9310 | 9.2950 | 0.0079 |
| TRPO-IPO | 2.5561 | 9.0792 | 0.0071 |
| TRPO-FAC | 10.0730 | 8.3481 | 0.0068 |
| CPO | **11.9573** | **6.0618** | 0.0074 |
| PCPO | 11.6731 | 6.8256 | 0.0074 |

Goal_Hopper_8Hazards

| Algorithm | $\bar{J}_r$ | $\bar{M}_c$ | $\bar{\rho}_c$ |
|---|---|---|---|
| TRPO | **32.8406** | 7.3477 | 0.0082 |
| TRPO-Lagrangian | 24.2180 | 6.4342 | **0.0069** |
| TRPO-SL | 26.1236 | 8.9366 | 0.0098 |
| TRPO-USL | 32.5692 | 8.1526 | 0.0080 |
| TRPO-IPO | 4.0118 | 7.2667 | 0.0082 |
| TRPO-FAC | 28.1388 | **6.3430** | 0.0076 |
| CPO | 27.2544 | 8.0783 | 0.0076 |
| PCPO | 30.7637 | 6.4343 | 0.0076 |

Goal_Arm3_8Hazards

| Algorithm | $\bar{J}_r$ | $\bar{M}_c$ | $\bar{\rho}_c$ |
|---|---|---|---|
| TRPO | 19.8716 | 23.8574 | 0.0293 |
| TRPO-Lagrangian | 6.0512 | 2.1411 | **0.0057** |
| TRPO-SL | 4.2161 | **0.4820** | 0.0115 |
| TRPO-USL | 15.6522 | 8.6754 | 0.0163 |
| TRPO-IPO | 2.4211 | 12.5567 | 0.0199 |
| TRPO-FAC | 10.0948 | 3.3072 | 0.0085 |
| CPO | 16.2682 | 22.1031 | 0.0210 |
| PCPO | **21.5110** | 16.2963 | 0.0211 |

Goal_Arm6_8Hazards

| Algorithm | $\bar{J}_r$ | $\bar{M}_c$ | $\bar{\rho}_c$ |
|---|---|---|---|
| TRPO | 4.3703 | 15.0087 | 0.0206 |
| TRPO-Lagrangian | 1.2386 | 6.8767 | **0.0107** |
| TRPO-SL | 2.1136 | 14.1806 | 0.0136 |
| TRPO-USL | 2.5704 | 9.4493 | 0.0186 |
| TRPO-IPO | 0.8242 | **5.5569** | 0.0129 |
| TRPO-FAC | 2.4243 | 8.9828 | 0.0124 |
| CPO | **4.3885** | 13.0115 | 0.0171 |
| PCPO | 1.1528 | 13.8961 | 0.0141 |

Goal_Drone_8Hazards

| Algorithm | $\bar{J}_r$ | $\bar{M}_c$ | $\bar{\rho}_c$ |
|---|---|---|---|
| TRPO | **19.6492** | 1.6839 | 0.0012 |
| TRPO-Lagrangian | 17.5182 | 1.0479 | 0.0010 |
| TRPO-SL | 11.0012 | **0.2030** | 0.0004 |
| TRPO-USL | 17.3535 | 1.1217 | 0.0008 |
| TRPO-IPO | 15.7189 | 0.8852 | 0.0007 |
| TRPO-FAC | 17.0156 | 1.0926 | 0.0005 |
| CPO | 18.3672 | 1.0204 | 0.0010 |
| PCPO | 5.0076 | 0.2334 | **0.0003** |

Table 22: Metrics of nine **Goal_{Robot}_8Ghosts** environments obtained from the final epoch.

Goal_Point_8Ghosts

| Algorithm | $\bar{J}_r$ | $\bar{M}_c$ | $\bar{\rho}_c$ |
|---|---|---|---|
| TRPO | 26.0478 | 6.8329 | 0.0073 |
| TRPO-Lagrangian | 26.3260 | **2.1498** | 0.0034 |
| TRPO-SL | 16.6548 | 4.0515 | 0.0058 |
| TRPO-USL | 22.1795 | 5.8895 | 0.0059 |
| TRPO-IPO | 20.1808 | 4.1169 | 0.0050 |
| TRPO-FAC | 25.9489 | 2.5654 | 0.0036 |
| CPO | **26.5064** | 2.6248 | **0.0034** |
| PCPO | 25.9672 | 3.8589 | 0.0054 |

Goal_Swimmer_8Ghosts

| Algorithm | $\bar{J}_r$ | $\bar{M}_c$ | $\bar{\rho}_c$ |
|---|---|---|---|
| TRPO | **30.3401** | 13.5808 | 0.0119 |
| TRPO-Lagrangian | 15.9952 | **2.1046** | **0.0061** |
| TRPO-SL | 7.8773 | 7.6875 | 0.0079 |
| TRPO-USL | 30.1229 | 8.9488 | 0.0105 |
| TRPO-IPO | 9.8646 | 10.0275 | 0.0091 |
| TRPO-FAC | 18.9950 | 4.4988 | 0.0069 |
| CPO | 26.6953 | 9.5202 | 0.0092 |
| PCPO | 26.2737 | 10.2204 | 0.0101 |

Goal_Ant_8Ghosts

| Algorithm | $\bar{J}_r$ | $\bar{M}_c$ | $\bar{\rho}_c$ |
|---|---|---|---|
| TRPO | 59.6760 | 10.3785 | 0.0099 |
| TRPO-Lagrangian | 28.5846 | **2.9654** | **0.0060** |
| TRPO-SL | 30.7285 | 41.2262 | 0.0342 |
| TRPO-USL | **61.2725** | 8.9165 | 0.0097 |
| TRPO-IPO | 2.9659 | 8.0972 | 0.0064 |
| TRPO-FAC | 44.2423 | 5.6508 | 0.0074 |
| CPO | 56.3422 | 9.8690 | 0.0095 |
| PCPO | 58.4684 | 9.8173 | 0.0095 |

Goal_Walker_8Ghosts

| Algorithm | $\bar{J}_r$ | $\bar{M}_c$ | $\bar{\rho}_c$ |
|---|---|---|---|
| TRPO | **63.2017** | 9.8771 | 0.0112 |
| TRPO-Lagrangian | 33.2534 | **2.5072** | **0.0054** |
| TRPO-SL | 37.8968 | 20.3758 | 0.0147 |
| TRPO-USL | 61.4547 | 9.6043 | 0.0105 |
| TRPO-IPO | 7.4640 | 9.1178 | 0.0080 |
| TRPO-FAC | 45.0094 | 4.9375 | 0.0071 |
| CPO | 60.1257 | 9.2117 | 0.0097 |
| PCPO | 43.8760 | 9.2932 | 0.0085 |

Goal_Humanoid_8Ghosts

| Algorithm | $\bar{J}_r$ | $\bar{M}_c$ | $\bar{\rho}_c$ |
|---|---|---|---|
| TRPO | 11.1891 | 9.9692 | 0.0098 |
| TRPO-Lagrangian | 5.0070 | **6.6812** | 0.0076 |
| TRPO-SL | 8.8939 | 17.0632 | 0.0107 |
| TRPO-USL | 10.6905 | 9.6248 | 0.0095 |
| TRPO-IPO | 1.0404 | 8.4966 | **0.0073** |
| TRPO-FAC | 9.2134 | 10.0716 | 0.0084 |
| CPO | 10.0778 | 10.3074 | 0.0092 |
| PCPO | **11.5003** | 9.0205 | 0.0093 |

Goal_Hopper_8Ghosts

| Algorithm | $\bar{J}_r$ | $\bar{M}_c$ | $\bar{\rho}_c$ |
|---|---|---|---|
| TRPO | **31.6643** | 8.1599 | 0.0100 |
| TRPO-Lagrangian | 14.1699 | **4.4744** | **0.0070** |
| TRPO-SL | 21.7761 | 12.4810 | 0.0122 |
| TRPO-USL | 31.2864 | 8.4550 | 0.0097 |
| TRPO-IPO | 5.4826 | 12.0015 | 0.0082 |
| TRPO-FAC | 28.8157 | 7.5453 | 0.0087 |
| CPO | 29.0408 | 7.5681 | 0.0086 |
| PCPO | 29.0858 | 8.0181 | 0.0090 |

Goal_Arm3_8Ghosts

| Algorithm | $\bar{J}_r$ | $\bar{M}_c$ | $\bar{\rho}_c$ |
|---|---|---|---|
| TRPO | 94.6660 | 35.7460 | 0.0348 |
| TRPO-Lagrangian | 15.4898 | 7.5123 | **0.0058** |
| TRPO-SL | 18.1207 | 10.7580 | 0.0174 |
| TRPO-USL | 62.1624 | 14.0682 | 0.0223 |
| TRPO-IPO | 4.0235 | 10.5251 | 0.0160 |
| TRPO-FAC | 37.9750 | **6.9701** | 0.0073 |
| CPO | 114.8705 | 15.1904 | 0.0159 |
| PCPO | **126.4001** | 10.1913 | 0.0143 |

Goal_Arm6_8Ghosts

| Algorithm | $\bar{J}_r$ | $\bar{M}_c$ | $\bar{\rho}_c$ |
|---|---|---|---|
| TRPO | 1.0157 | 49.0135 | 0.0466 |
| TRPO-Lagrangian | 0.5470 | **8.4307** | 0.0190 |
| TRPO-SL | 0.6078 | 20.5269 | 0.0356 |
| TRPO-USL | 0.9856 | 41.7054 | 0.0427 |
| TRPO-IPO | 0.7336 | 12.4453 | 0.0233 |
| TRPO-FAC | 0.7861 | 9.4493 | 0.0170 |
| CPO | **9.9993** | 22.5031 | 0.0234 |
| PCPO | 0.8845 | 15.9718 | **0.0162** |

Goal_Drone_8Ghosts

| Algorithm | $\bar{J}_r$ | $\bar{M}_c$ | $\bar{\rho}_c$ |
|---|---|---|---|
| TRPO | 17.9484 | 1.7287 | 0.0011 |
| TRPO-Lagrangian | 18.9773 | 0.9218 | 0.0008 |
| TRPO-SL | 12.1413 | **0.2500** | 0.0004 |
| TRPO-USL | 10.7517 | 0.9741 | 0.0011 |
| TRPO-IPO | 11.5210 | 0.6817 | 0.0006 |
| TRPO-FAC | **20.1014** | 0.7630 | 0.0006 |
| CPO | 18.4723 | 1.2188 | 0.0008 |
| PCPO | 6.5276 | 0.3859 | **0.0003** |

Table 23: Metrics of nine **Push_{Robot}_8Hazards** environments obtained from the final epoch.

Push_Point_8Hazards

| Algorithm | $\bar{J}_r$ | $\bar{M}_c$ | $\bar{\rho}_c$ |
|---|---|---|---|
| TRPO | **11.3060** | 7.2536 | 0.0084 |
| TRPO-Lagrangian | 4.1189 | **1.8268** | **0.0037** |
| TRPO-SL | 3.0553 | 6.6139 | 0.0058 |
| TRPO-USL | 9.1904 | 6.6179 | 0.0064 |
| TRPO-IPO | 1.3370 | 4.0476 | 0.0051 |
| TRPO-FAC | 6.0431 | 2.1250 | 0.0039 |
| CPO | 9.7522 | 5.6406 | 0.0066 |
| PCPO | 9.1434 | 6.5665 | 0.0066 |

Push_Swimmer_8Hazards

| Algorithm | $\bar{J}_r$ | $\bar{M}_c$ | $\bar{\rho}_c$ |
|---|---|---|---|
| TRPO | **86.1557** | 11.9235 | 0.0102 |
| TRPO-Lagrangian | 52.0782 | **4.5645** | 0.0070 |
| TRPO-SL | 13.1869 | 7.7554 | **0.0057** |
| TRPO-USL | 64.0705 | 9.4963 | 0.0085 |
| TRPO-IPO | 6.3843 | 8.4329 | 0.0077 |
| TRPO-FAC | 48.2986 | 5.8675 | 0.0064 |
| CPO | 57.4370 | 6.9551 | 0.0072 |
| PCPO | 56.2598 | 6.1634 | 0.0076 |

Push_Ant_8Hazards

| Algorithm | $\bar{J}_r$ | $\bar{M}_c$ | $\bar{\rho}_c$ |
|---|---|---|---|
| TRPO | **13.4378** | 9.4740 | 0.0091 |
| TRPO-Lagrangian | 1.1582 | **1.5948** | **0.0043** |
| TRPO-SL | 3.5622 | 47.7602 | 0.0217 |
| TRPO-USL | 11.2763 | 9.3930 | 0.0086 |
| TRPO-IPO | 1.1986 | 5.9120 | 0.0061 |
| TRPO-FAC | 2.5905 | 2.7927 | 0.0050 |
| CPO | 12.7081 | 7.5742 | 0.0082 |
| PCPO | 11.0161 | 8.7780 | 0.0087 |

Push_Walker_8Hazards

| Algorithm | $\bar{J}_r$ | $\bar{M}_c$ | $\bar{\rho}_c$ |
|---|---|---|---|
| TRPO | **5.0574** | 10.8840 | 0.0089 |
| TRPO-Lagrangian | 1.5035 | **2.4237** | **0.0040** |
| TRPO-SL | 1.7263 | 17.5680 | 0.0082 |
| TRPO-USL | 2.8786 | 9.3900 | 0.0078 |
| TRPO-IPO | 0.7991 | 3.6377 | 0.0070 |
| TRPO-FAC | 1.5393 | 3.2465 | 0.0047 |
| CPO | 4.3412 | 7.8450 | 0.0075 |
| PCPO | 1.1548 | 9.2470 | 0.0075 |

Push_Humanoid_8Hazards

| Algorithm | $\bar{J}_r$ | $\bar{M}_c$ | $\bar{\rho}_c$ |
|---|---|---|---|
| TRPO | 0.9545 | 10.6542 | 0.0096 |
| TRPO-Lagrangian | 0.7407 | 3.1758 | **0.0062** |
| TRPO-SL | 0.2992 | 9.0239 | 0.0092 |
| TRPO-USL | 0.8102 | 7.3410 | 0.0093 |
| TRPO-IPO | 0.8194 | 6.0952 | 0.0074 |
| TRPO-FAC | 0.9641 | **3.0034** | 0.0068 |
| CPO | 0.8147 | 8.6884 | 0.0080 |
| PCPO | **1.0445** | 8.1230 | 0.0084 |

Push_Hopper_8Hazards

| Algorithm | $\bar{J}_r$ | $\bar{M}_c$ | $\bar{\rho}_c$ |
|---|---|---|---|
| TRPO | **3.6134** | 10.3693 | 0.0095 |
| TRPO-Lagrangian | 0.8384 | **2.0782** | 0.0052 |
| TRPO-SL | 1.5115 | 8.2643 | 0.0080 |
| TRPO-USL | 2.3949 | 11.2835 | 0.0088 |
| TRPO-IPO | 0.3718 | 7.4184 | 0.0083 |
| TRPO-FAC | 1.0928 | 3.8033 | 0.0069 |
| CPO | 2.3108 | 11.2012 | 0.0082 |
| PCPO | 0.9565 | 8.8373 | 0.0083 |

Push_Arm3_8Hazards

| Algorithm | $\bar{J}_r$ | $\bar{M}_c$ | $\bar{\rho}_c$ |
|---|---|---|---|
| TRPO | 0.0438 | 37.7114 | 0.0414 |
| TRPO-Lagrangian | -194.8455 | **2.7071** | **0.0062** |
| TRPO-SL | **0.0906** | 7.3980 | 0.0176 |
| TRPO-USL | -42.2457 | 10.6065 | 0.0189 |
| TRPO-IPO | -420.0890 | 25.0669 | 0.0224 |
| TRPO-FAC | -114.8912 | 7.8944 | 0.0086 |
| CPO | 0.0249 | 11.3773 | 0.0128 |
| PCPO | -30.9294 | 10.4467 | 0.0207 |

Push_Arm6_8Hazards

| Algorithm | $\bar{J}_r$ | $\bar{M}_c$ | $\bar{\rho}_c$ |
|---|---|---|---|
| TRPO | 1.1128 | 15.9080 | 0.0190 |
| TRPO-Lagrangian | 0.9490 | 7.1961 | 0.0110 |
| TRPO-SL | -220.2115 | 38.7175 | 0.0144 |
| TRPO-USL | -0.6530 | 16.7103 | 0.0182 |
| TRPO-IPO | 1.1291 | 8.3642 | 0.0113 |
| TRPO-FAC | 1.0648 | 9.4750 | 0.0152 |
| CPO | **1.1699** | **6.6375** | **0.0103** |
| PCPO | 1.1459 | 10.0104 | 0.0112 |

Push_Drone_8Hazards

| Algorithm | $\bar{J}_r$ | $\bar{M}_c$ | $\bar{\rho}_c$ |
|---|---|---|---|
| TRPO | 0.9332 | 0.3324 | 0.0002 |
| TRPO-Lagrangian | 1.0967 | 0.3197 | 0.0003 |
| TRPO-SL | 1.0154 | 0.0783 | 0.0001 |
| TRPO-USL | 0.9410 | 0.0996 | 0.0001 |
| TRPO-IPO | 1.0394 | 0.4229 | 0.0002 |
| TRPO-FAC | 1.0820 | 0.2380 | 0.0002 |
| CPO | **1.1261** | 0.2409 | 0.0003 |
| PCPO | 0.9844 | **0.0049** | **0.0001** |

Table 24: Metrics of nine **Chase_{Robot}_8Hazards** environments obtained from the final epoch.

Chase_Point_8Hazards

| Algorithm | $\bar{J}_r$ | $\bar{M}_c$ | $\bar{\rho}_c$ |
|---|---|---|---|
| TRPO | **1.3122** | 3.5553 | 0.0068 |
| TRPO-Lagrangian | 1.0879 | **2.8816** | **0.0046** |
| TRPO-SL | 0.8385 | 5.6000 | 0.0058 |
| TRPO-USL | 1.1433 | 5.7574 | 0.0080 |
| TRPO-IPO | 0.7959 | 8.5632 | 0.0061 |
| TRPO-FAC | 1.0333 | 3.0887 | 0.0053 |
| CPO | 1.2897 | 5.0677 | 0.0063 |
| PCPO | 1.0035 | 7.9018 | 0.0084 |

Chase_Swimmer_8Hazards

| Algorithm | $\bar{J}_r$ | $\bar{M}_c$ | $\bar{\rho}_c$ |
|---|---|---|---|
| TRPO | 1.2491 | 7.0269 | 0.0100 |
| TRPO-Lagrangian | -0.2346 | **4.8860** | **0.0058** |
| TRPO-SL | 0.0518 | 9.2681 | 0.0071 |
| TRPO-USL | 1.2227 | 9.2911 | 0.0103 |
| TRPO-IPO | -1.0848 | 10.5546 | 0.0080 |
| TRPO-FAC | 0.6411 | 9.1446 | 0.0078 |
| CPO | **1.2540** | 8.1671 | 0.0082 |
| PCPO | 1.2152 | 8.2717 | 0.0090 |

Chase_Ant_8Hazards

| Algorithm | $\bar{J}_r$ | $\bar{M}_c$ | $\bar{\rho}_c$ |
|---|---|---|---|
| TRPO | 1.3504 | 6.1101 | 0.0106 |
| TRPO-Lagrangian | -0.3563 | 2.5016 | **0.0040** |
| TRPO-SL | 0.7921 | 16.9846 | 0.0222 |
| TRPO-USL | 1.3841 | 8.0640 | 0.0096 |
| TRPO-IPO | -0.9314 | 2.5529 | 0.0048 |
| TRPO-FAC | -0.0258 | 3.5439 | 0.0048 |
| CPO | **1.4104** | 5.7863 | 0.0087 |
| PCPO | 1.3122 | 6.9139 | 0.0097 |

Chase_Walker_8Hazards

| Algorithm | $\bar{J}_r$ | $\bar{M}_c$ | $\bar{\rho}_c$ |
|---|---|---|---|
| TRPO | 0.4890 | 7.6845 | 0.0088 |
| TRPO-Lagrangian | -0.0922 | 2.5167 | 0.0045 |
| TRPO-SL | -0.2116 | 10.7167 | 0.0094 |
| TRPO-USL | 0.4639 | 7.7035 | 0.0082 |
| TRPO-IPO | -0.8223 | **2.3954** | **0.0038** |
| TRPO-FAC | -0.0368 | 2.7105 | 0.0047 |
| CPO | **0.7406** | 10.4993 | 0.0086 |
| PCPO | 0.6347 | 8.8652 | 0.0080 |

Chase_Humanoid_8Hazards

| Algorithm | $\bar{J}_r$ | $\bar{M}_c$ | $\bar{\rho}_c$ |
|---|---|---|---|
| TRPO | **0.2330** | 12.1455 | 0.0152 |
| TRPO-Lagrangian | -0.6855 | **3.4234** | **0.0047** |
| TRPO-SL | -0.2271 | 11.8001 | 0.0121 |
| TRPO-USL | -0.1503 | 18.6011 | 0.0149 |
| TRPO-IPO | -0.8074 | 6.4163 | 0.0054 |
| TRPO-FAC | -0.5826 | 3.6663 | 0.0050 |
| CPO | -0.3322 | 12.1665 | 0.0109 |
| PCPO | -0.0971 | 10.3441 | 0.0113 |

Chase_Hopper_8Hazards

| Algorithm | $\bar{J}_r$ | $\bar{M}_c$ | $\bar{\rho}_c$ |
|---|---|---|---|
| TRPO | **0.6099** | 12.1675 | 0.0134 |
| TRPO-Lagrangian | -0.3641 | **3.2170** | **0.0039** |
| TRPO-SL | 0.4957 | 5.4355 | 0.0089 |
| TRPO-USL | 0.4819 | 11.0919 | 0.0123 |
| TRPO-IPO | -0.7766 | 6.1236 | 0.0061 |
| TRPO-FAC | -0.3651 | 3.7391 | 0.0055 |
| CPO | 0.4829 | 6.7117 | 0.0083 |
| PCPO | -0.1457 | 7.4290 | 0.0068 |

Chase_Arm3_8Hazards

| Algorithm | $\bar{J}_r$ | $\bar{M}_c$ | $\bar{\rho}_c$ |
|---|---|---|---|
| TRPO | 0.7772 | 20.6230 | 0.0312 |
| TRPO-Lagrangian | -0.2739 | 5.1692 | **0.0079** |
| TRPO-SL | 0.0007 | 4.2869 | 0.0142 |
| TRPO-USL | 0.7825 | 14.1736 | 0.0284 |
| TRPO-IPO | -0.4137 | 10.6685 | 0.0223 |
| TRPO-FAC | 0.3648 | **3.3449** | 0.0127 |
| CPO | **0.8051** | 17.4917 | 0.0252 |
| PCPO | 0.7355 | 25.8202 | 0.0291 |

Chase_Arm6_8Hazards

| Algorithm | $\bar{J}_r$ | $\bar{M}_c$ | $\bar{\rho}_c$ |
|---|---|---|---|
| TRPO | -0.3969 | 60.5704 | 0.0598 |
| TRPO-Lagrangian | -0.4860 | **2.4602** | **0.0075** |
| TRPO-SL | -0.5420 | 12.1256 | 0.0237 |
| TRPO-USL | -0.5734 | 53.4455 | 0.0575 |
| TRPO-IPO | **-0.2855** | 11.6769 | 0.0085 |
| TRPO-FAC | -0.3083 | 13.2429 | 0.0263 |
| CPO | -0.3278 | 16.9609 | 0.0247 |
| PCPO | -0.2883 | 45.6164 | 0.0463 |

Chase_Drone_8Hazards

| Algorithm | $\bar{J}_r$ | $\bar{M}_c$ | $\bar{\rho}_c$ |
|---|---|---|---|
| TRPO | 1.0351 | 0.6939 | 0.0008 |
| TRPO-Lagrangian | 0.8211 | 1.3456 | 0.0008 |
| TRPO-SL | -1.3055 | 0.2603 | **0.0002** |
| TRPO-USL | 0.7461 | 1.2159 | 0.0006 |
| TRPO-IPO | 0.2518 | 0.5786 | 0.0005 |
| TRPO-FAC | **1.1192** | 0.2374 | 0.0006 |
| CPO | 0.7682 | 0.9075 | 0.0006 |
| PCPO | 0.6172 | 0.6374 | 0.0012 |

Table 25: Metrics of nine **Defense_{Robot}_8Hazards** environments obtained from the final epoch.

Defense_Point_8Hazards

| Algorithm | $\bar{J}_r$ | $\bar{M}_c$ | $\bar{\rho}_c$ |
|---|---|---|---|
| TRPO | **71.7851** | 37.5050 | 0.0308 |
| TRPO-Lagrangian | -12.2159 | 1.1776 | 0.0026 |
| TRPO-SL | -89.8828 | 3.1691 | 0.0070 |
| TRPO-USL | -109.7828 | 9.9285 | 0.0086 |
| TRPO-IPO | -330.4252 | **0.7309** | 0.0035 |
| TRPO-FAC | -269.0397 | 0.7334 | **0.0015** |
| CPO | 36.7643 | 7.1534 | 0.0071 |
| PCPO | 19.0943 | 1.9388 | 0.0048 |

Defense_Swimmer_8Hazards

| Algorithm | $\bar{J}_r$ | $\bar{M}_c$ | $\bar{\rho}_c$ |
|---|---|---|---|
| TRPO | 119.9896 | 44.5965 | 0.0405 |
| TRPO-Lagrangian | -85.0177 | **0.2487** | **0.0031** |
| TRPO-SL | -41.8928 | 1.3295 | 0.0118 |
| TRPO-USL | **139.8915** | 13.5482 | 0.0150 |
| TRPO-IPO | -233.1962 | 7.6313 | 0.0070 |
| TRPO-FAC | -91.7454 | 0.8809 | 0.0032 |
| CPO | 34.3226 | 2.7346 | 0.0072 |
| PCPO | 91.1387 | 5.1068 | 0.0084 |

Defense_Ant_8Hazards

| Algorithm | $\bar{J}_r$ | $\bar{M}_c$ | $\bar{\rho}_c$ |
|---|---|---|---|
| TRPO | **65.9815** | 46.1871 | 0.0214 |
| TRPO-Lagrangian | -190.9671 | 1.5799 | **0.0040** |
| TRPO-SL | -15.0035 | 14.7914 | 0.0143 |
| TRPO-USL | -9.1186 | 25.8625 | 0.0126 |
| TRPO-IPO | -205.8713 | 6.0119 | 0.0044 |
| TRPO-FAC | -204.4595 | **1.4105** | 0.0041 |
| CPO | -22.6369 | 17.9356 | 0.0132 |
| PCPO | -42.0119 | 17.1633 | 0.0120 |

Defense_Walker_8Hazards

| Algorithm | $\bar{J}_r$ | $\bar{M}_c$ | $\bar{\rho}_c$ |
|---|---|---|---|
| TRPO | **63.0381** | 52.1661 | 0.0326 |
| TRPO-Lagrangian | -221.9464 | **0.8080** | **0.0032** |
| TRPO-SL | -28.2392 | 21.2179 | 0.0142 |
| TRPO-USL | 19.2097 | 23.4844 | 0.0182 |
| TRPO-IPO | -213.4079 | 2.7606 | 0.0045 |
| TRPO-FAC | -183.6202 | 1.6905 | 0.0035 |
| CPO | 32.0705 | 14.7761 | 0.0151 |
| PCPO | 43.8441 | 17.0562 | 0.0161 |

Defense_Humanoid_8Hazards

| Algorithm | $\bar{J}_r$ | $\bar{M}_c$ | $\bar{\rho}_c$ |
|---|---|---|---|
| TRPO | -279.6928 | 4.3248 | 0.0042 |
| TRPO-Lagrangian | -287.5846 | 3.0248 | 0.0035 |
| TRPO-SL | -325.6846 | 2.0650 | 0.0039 |
| TRPO-USL | -318.2901 | 3.5935 | 0.0043 |
| TRPO-IPO | -281.2530 | 4.3968 | 0.0038 |
| TRPO-FAC | -271.4645 | **2.0044** | **0.0034** |
| CPO | **-246.6409** | 5.8980 | 0.0049 |
| PCPO | -317.0349 | 2.9953 | 0.0040 |

Defense_Hopper_8Hazards

| Algorithm | $\bar{J}_r$ | $\bar{M}_c$ | $\bar{\rho}_c$ |
|---|---|---|---|
| TRPO | -79.4386 | 26.9427 | 0.0202 |
| TRPO-Lagrangian | -304.2345 | 1.1963 | **0.0029** |
| TRPO-SL | -207.0506 | 8.9198 | 0.0138 |
| TRPO-USL | **57.7316** | 28.5037 | 0.0234 |
| TRPO-IPO | -248.0784 | 6.6735 | 0.0046 |
| TRPO-FAC | -233.1694 | **0.7496** | 0.0038 |
| CPO | -271.5419 | 8.3413 | 0.0077 |
| PCPO | -279.4999 | 7.2803 | 0.0077 |

Defense_Arm3_8Hazards

| Algorithm | $\bar{J}_r$ | $\bar{M}_c$ | $\bar{\rho}_c$ |
|---|---|---|---|
| TRPO | 169.5352 | 22.0750 | 0.0301 |
| TRPO-Lagrangian | 151.7291 | **0.7971** | **0.0056** |
| TRPO-SL | 112.3637 | 1.1085 | 0.0160 |
| TRPO-USL | 164.4992 | 5.3212 | 0.0163 |
| TRPO-IPO | 94.1636 | 9.1085 | 0.0171 |
| TRPO-FAC | **180.9871** | 1.7731 | 0.0064 |
| CPO | 167.4984 | 16.4595 | 0.0162 |
| PCPO | 160.2841 | 22.2282 | 0.0189 |

Defense_Arm6_8Hazards

| Algorithm | $\bar{J}_r$ | $\bar{M}_c$ | $\bar{\rho}_c$ |
|---|---|---|---|
| TRPO | **183.9203** | 56.5334 | 0.0548 |
| TRPO-Lagrangian | 169.9900 | **1.0108** | **0.0045** |
| TRPO-SL | 171.8430 | 13.3277 | 0.0229 |
| TRPO-USL | 183.7060 | 52.3346 | 0.0528 |
| TRPO-IPO | 127.3447 | 3.8719 | 0.0051 |
| TRPO-FAC | 175.8257 | 2.3101 | 0.0109 |
| CPO | 174.7701 | 22.8158 | 0.0346 |
| PCPO | 174.4207 | 30.1276 | 0.0264 |

Defense_Drone_8Hazards

| Algorithm | $\bar{J}_r$ | $\bar{M}_c$ | $\bar{\rho}_c$ |
|---|---|---|---|
| TRPO | -241.5720 | 0.0771 | 0.0002 |
| TRPO-Lagrangian | -245.7311 | 0.2276 | 0.0002 |
| TRPO-SL | -371.7727 | **0.0000** | **0.0001** |
| TRPO-USL | -336.7727 | 0.2161 | 0.0002 |
| TRPO-IPO | -275.5550 | 0.2600 | 0.0002 |
| TRPO-FAC | -215.4844 | 0.0691 | 0.0001 |
| CPO | **-212.1858** | 0.0236 | 0.0002 |
| PCPO | -219.4308 | 0.3358 | 0.0003 |

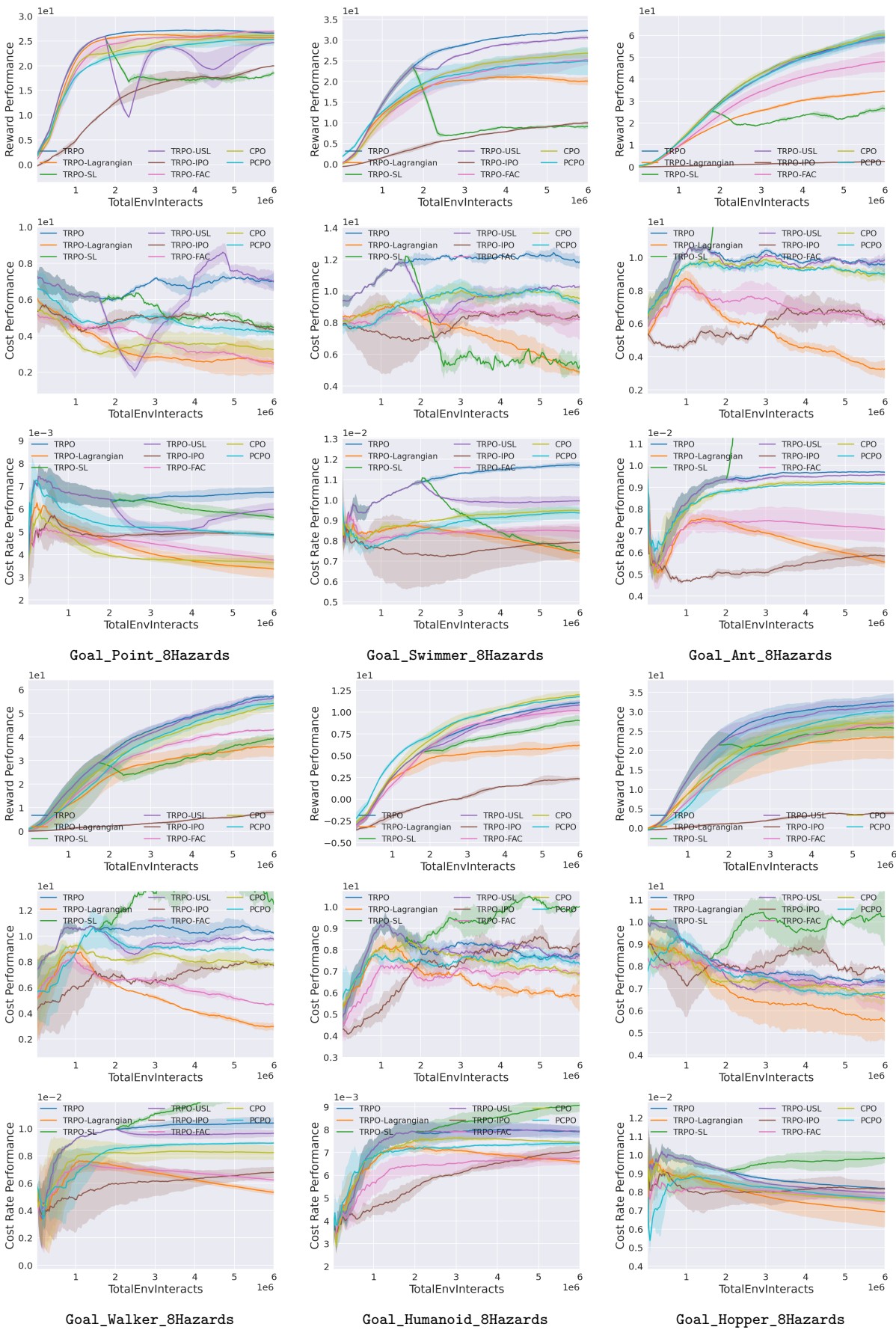

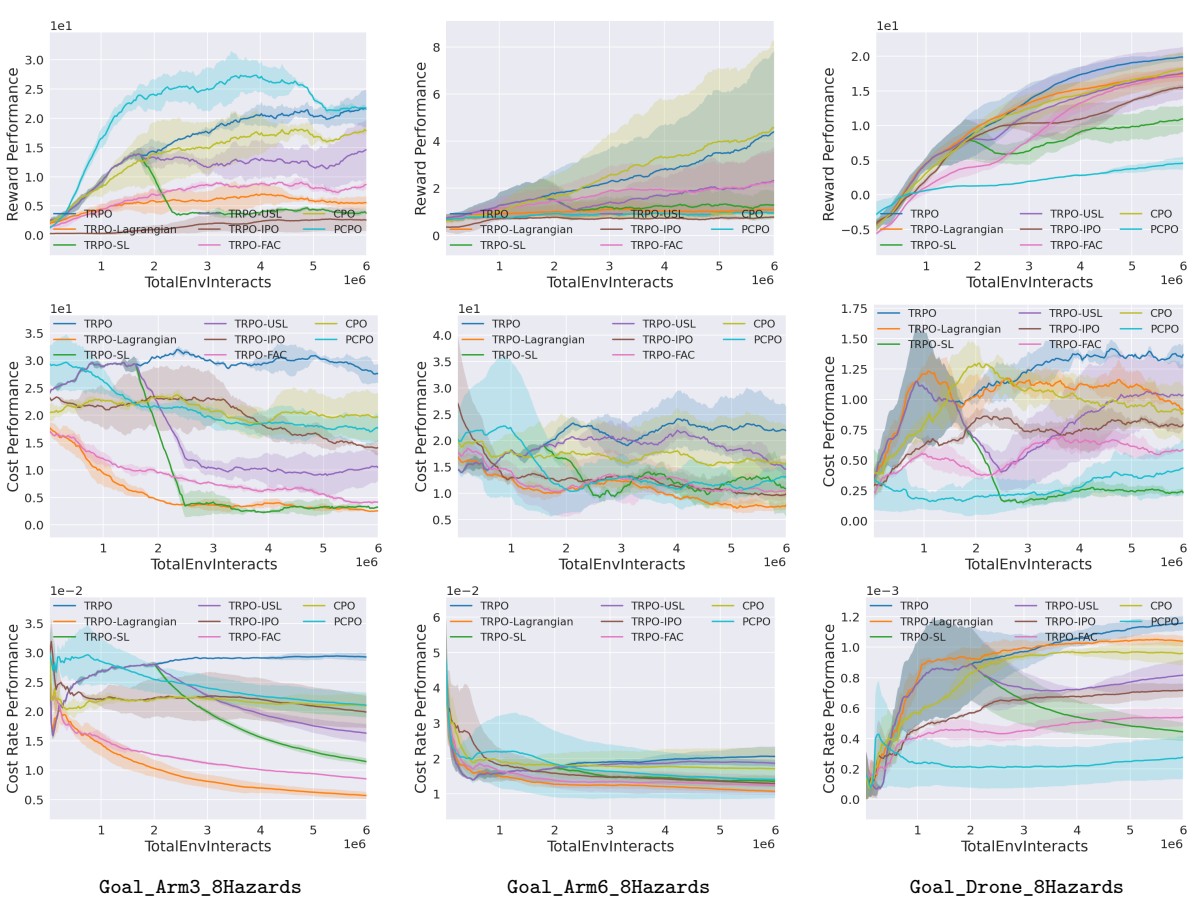

Figure 14: `Goal_{Robot}_8Hazards`

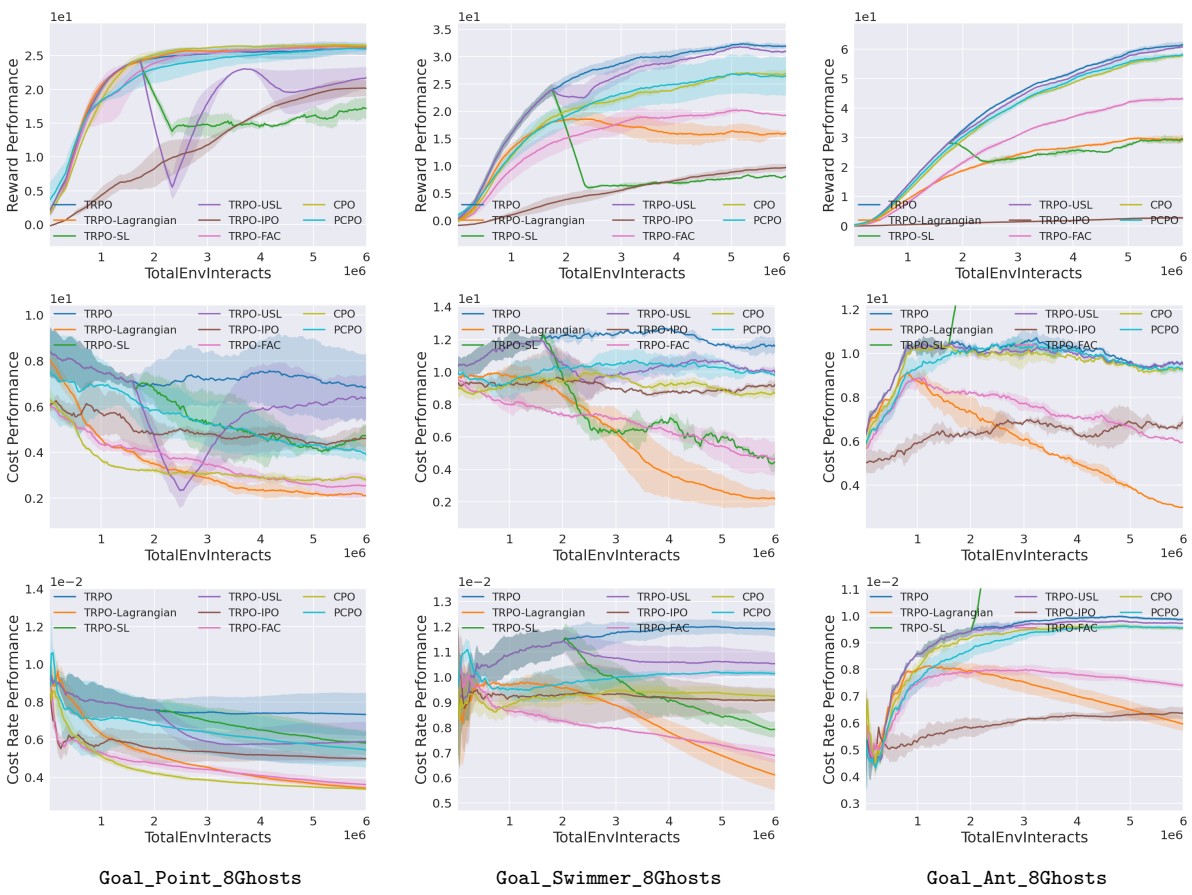

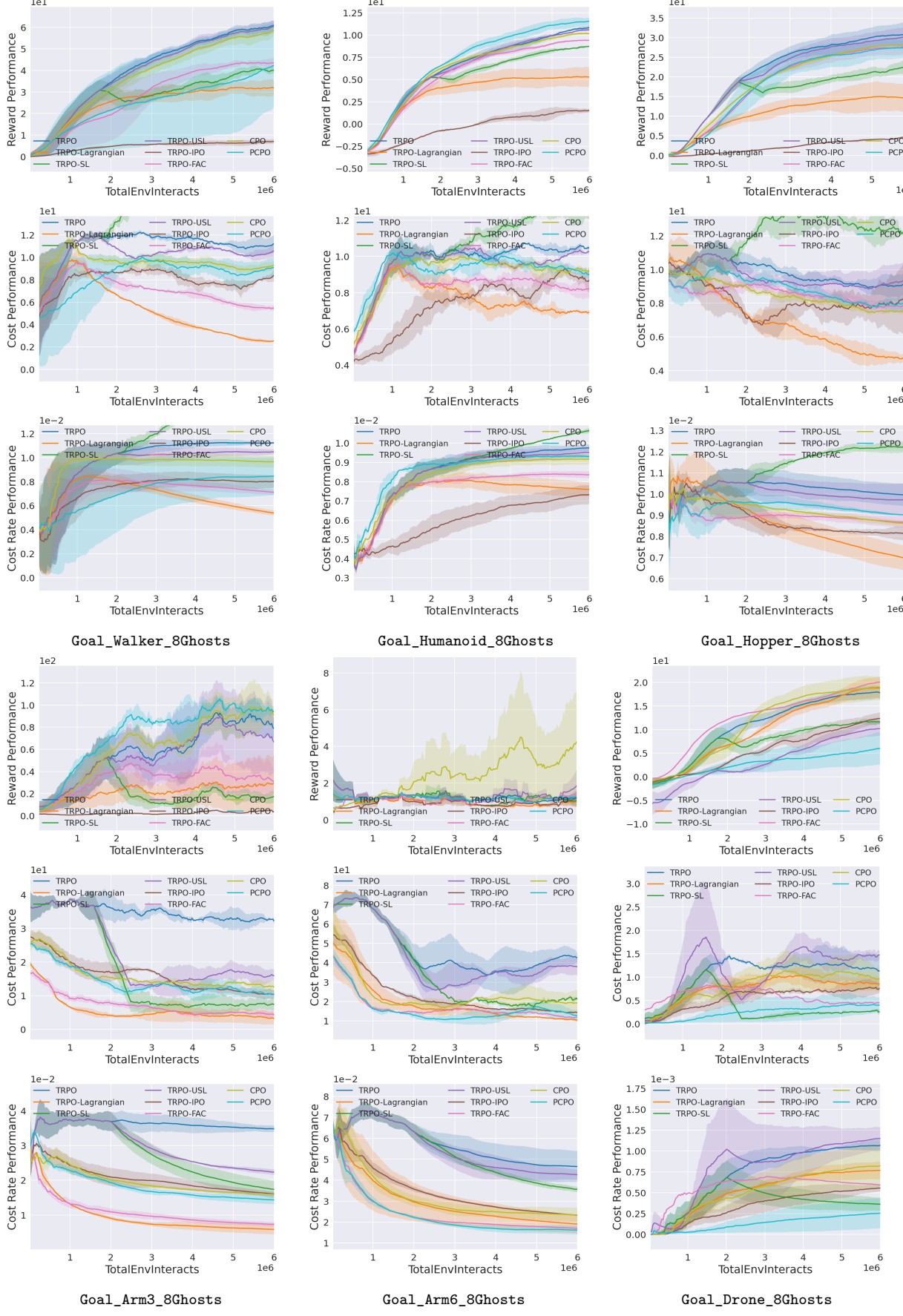

Figure 15: `Goal_{Robot}_8Ghosts`

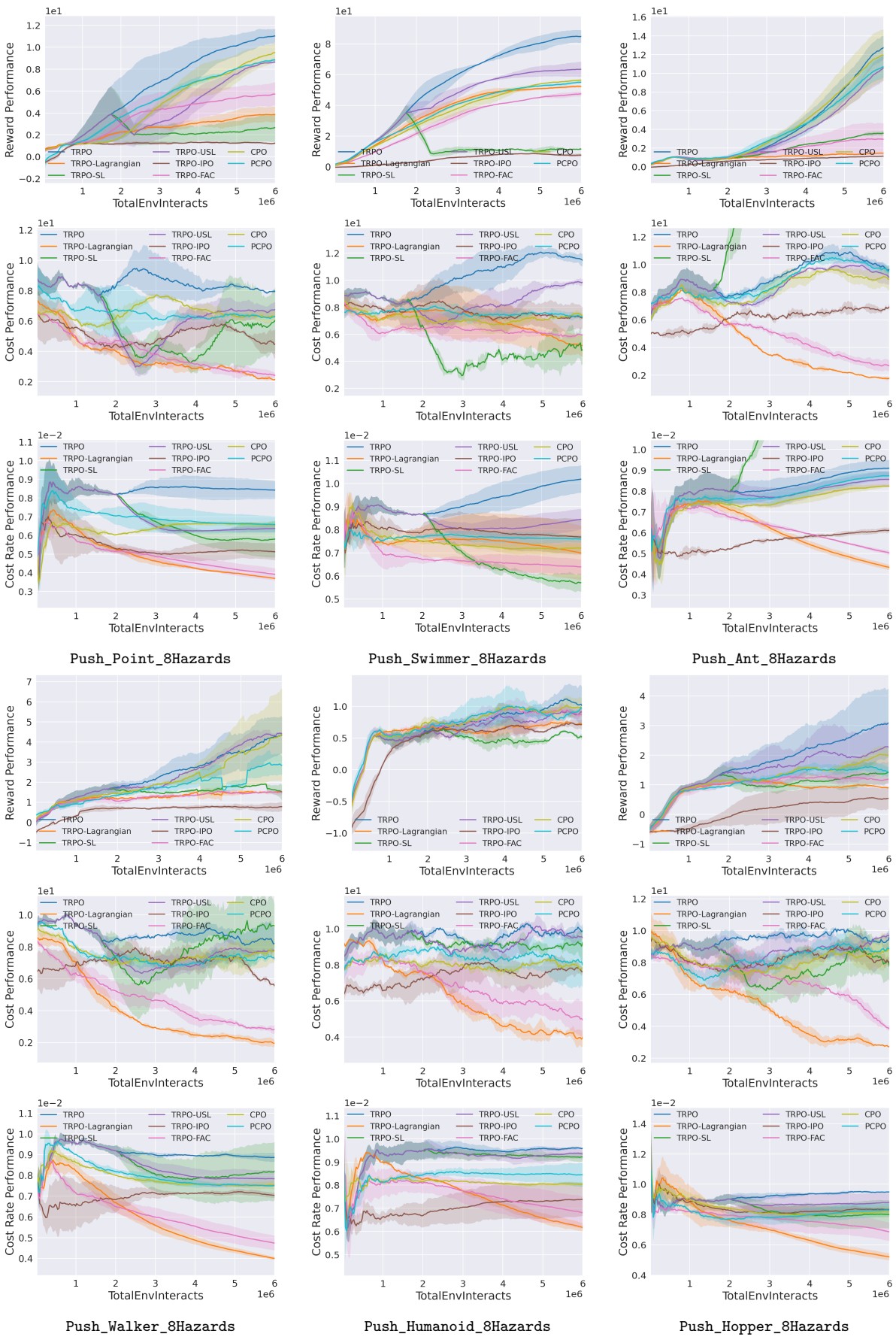

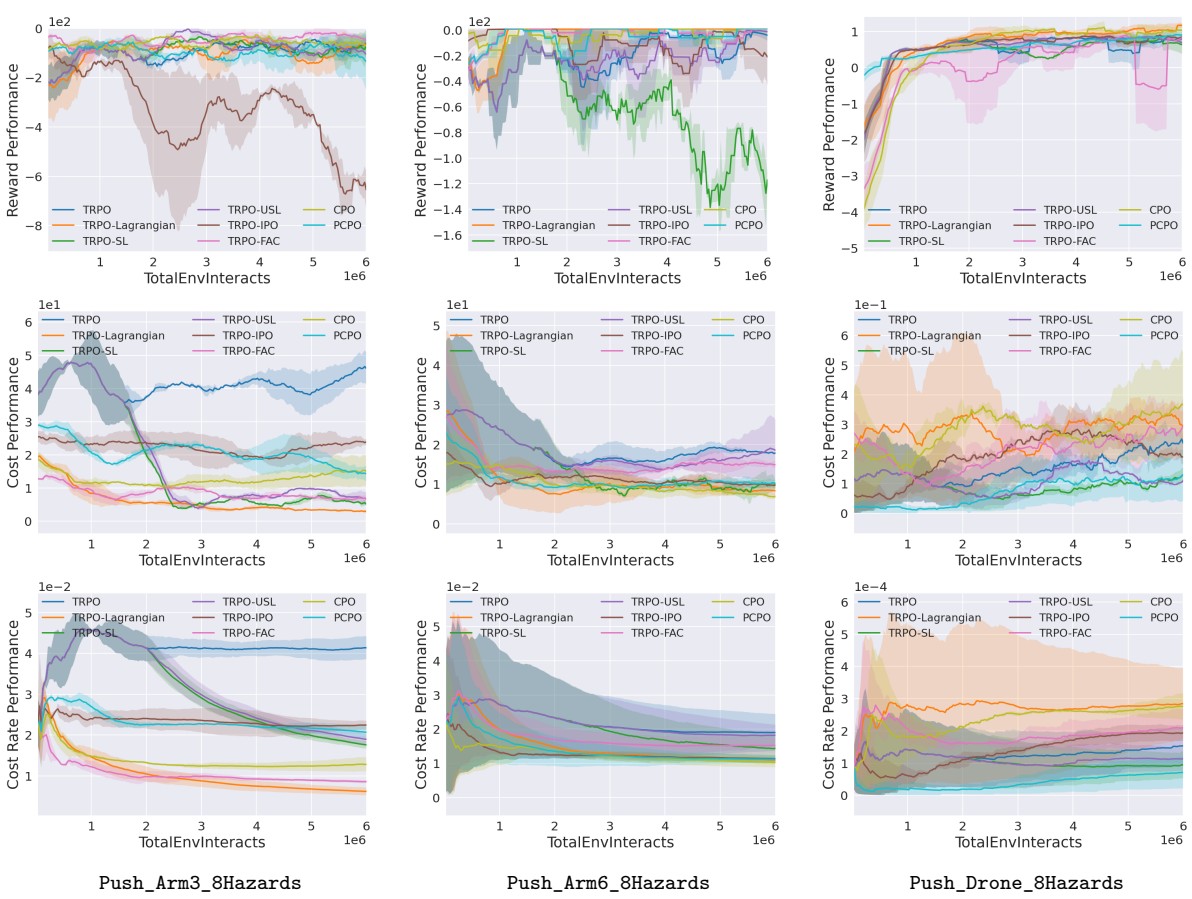

Push_Arm3_8Hazards      Push_Arm6_8Hazards      Push_Drone_8Hazards

Figure 16: Push_{Robot}_8Hazards

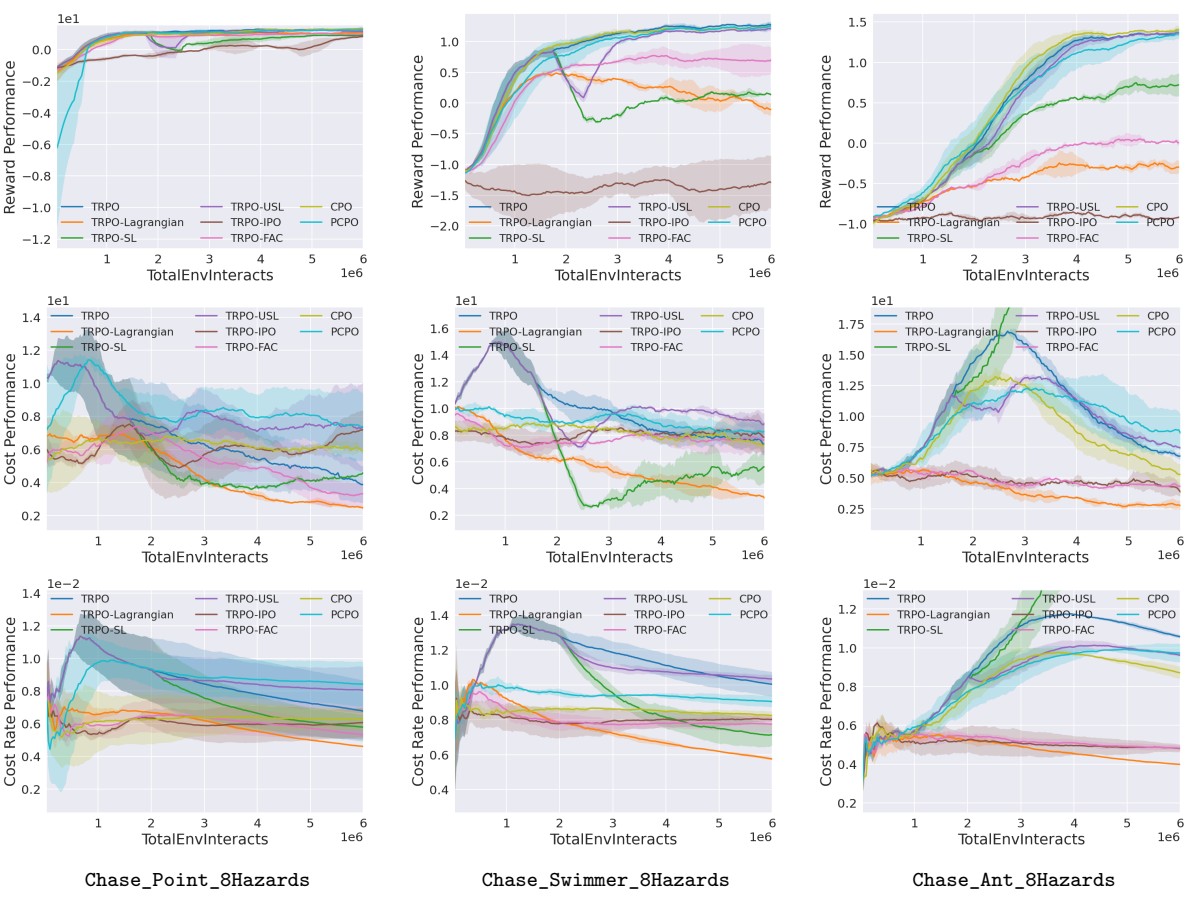

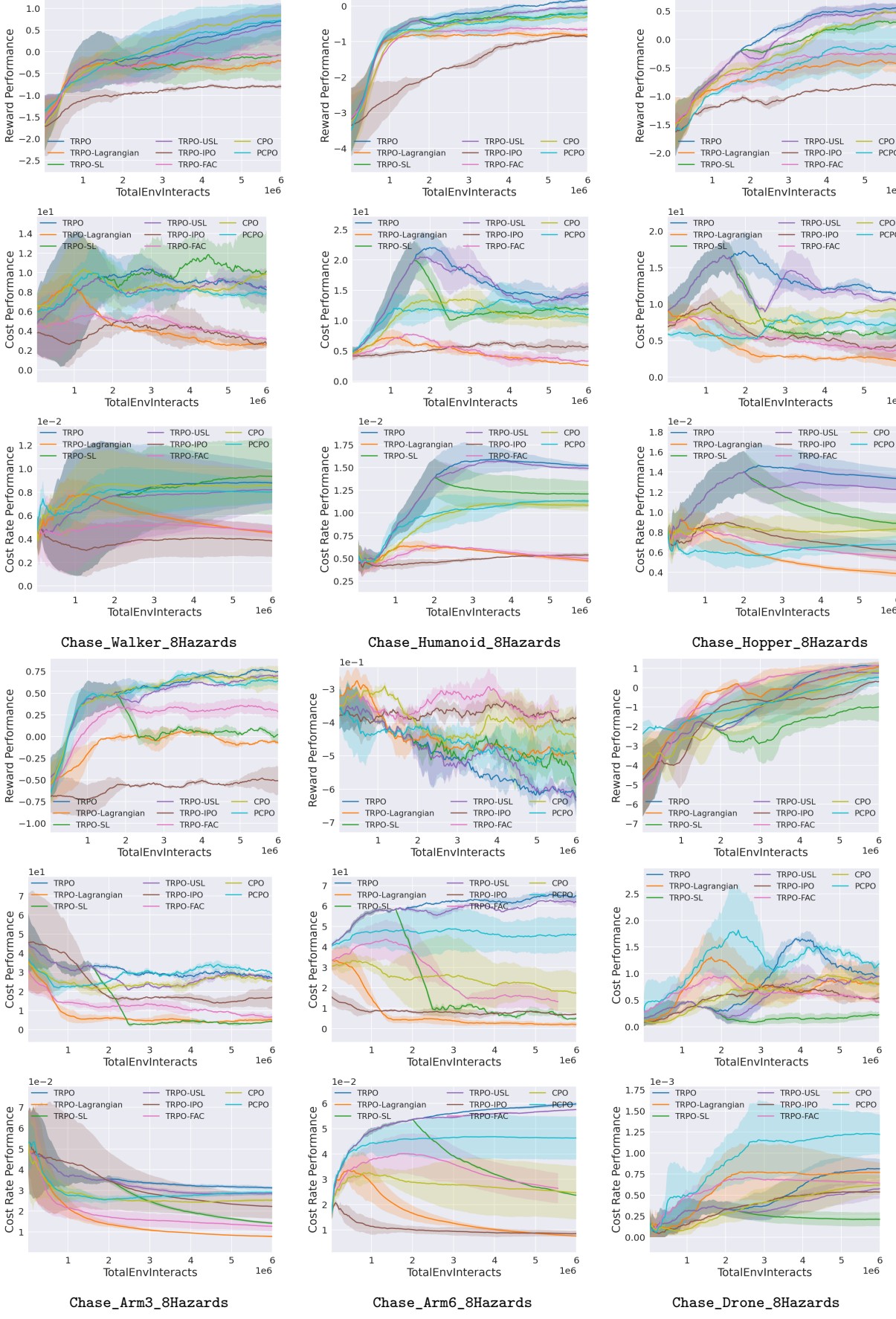

Figure 17: Chase_{Robot}_8Hazards

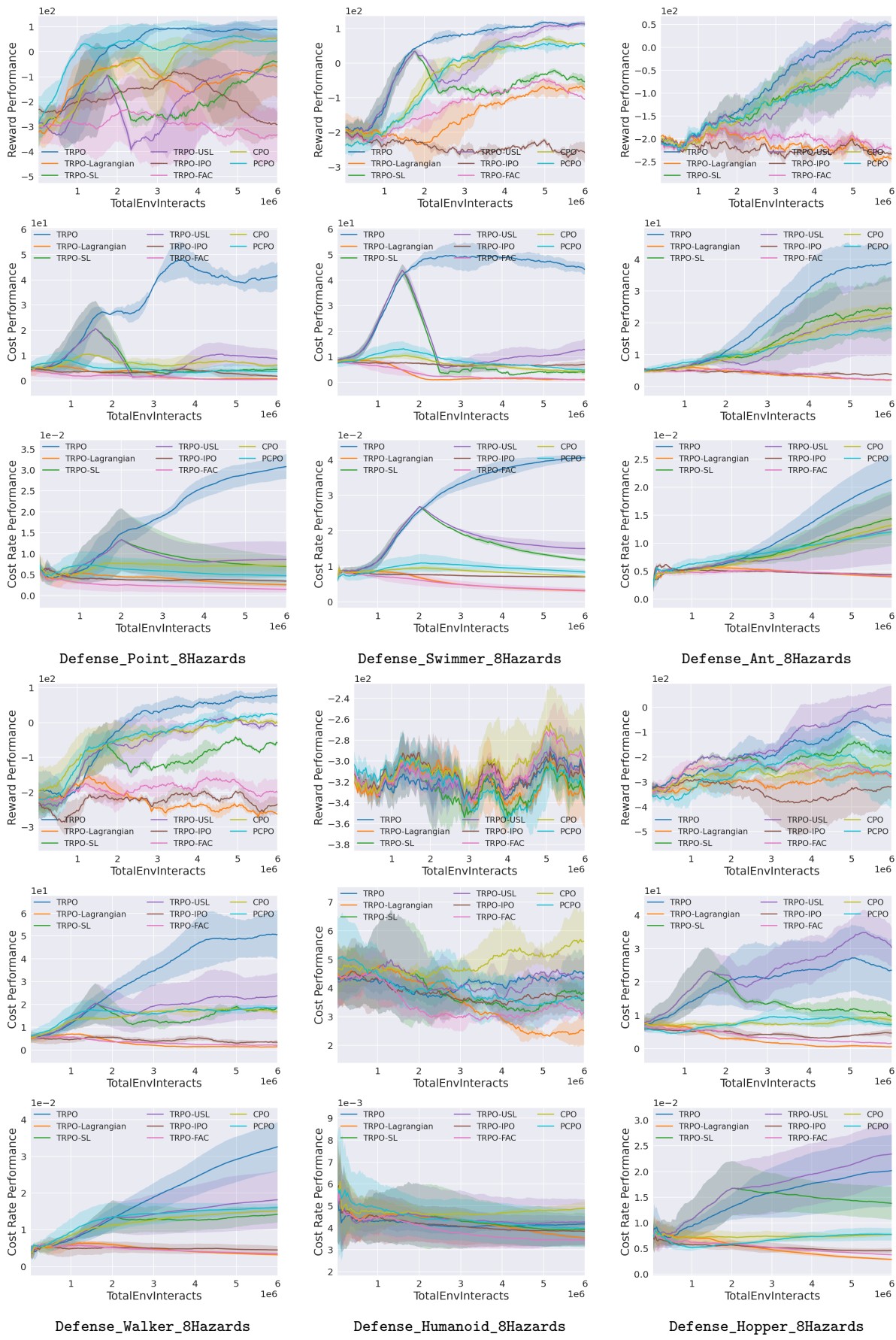

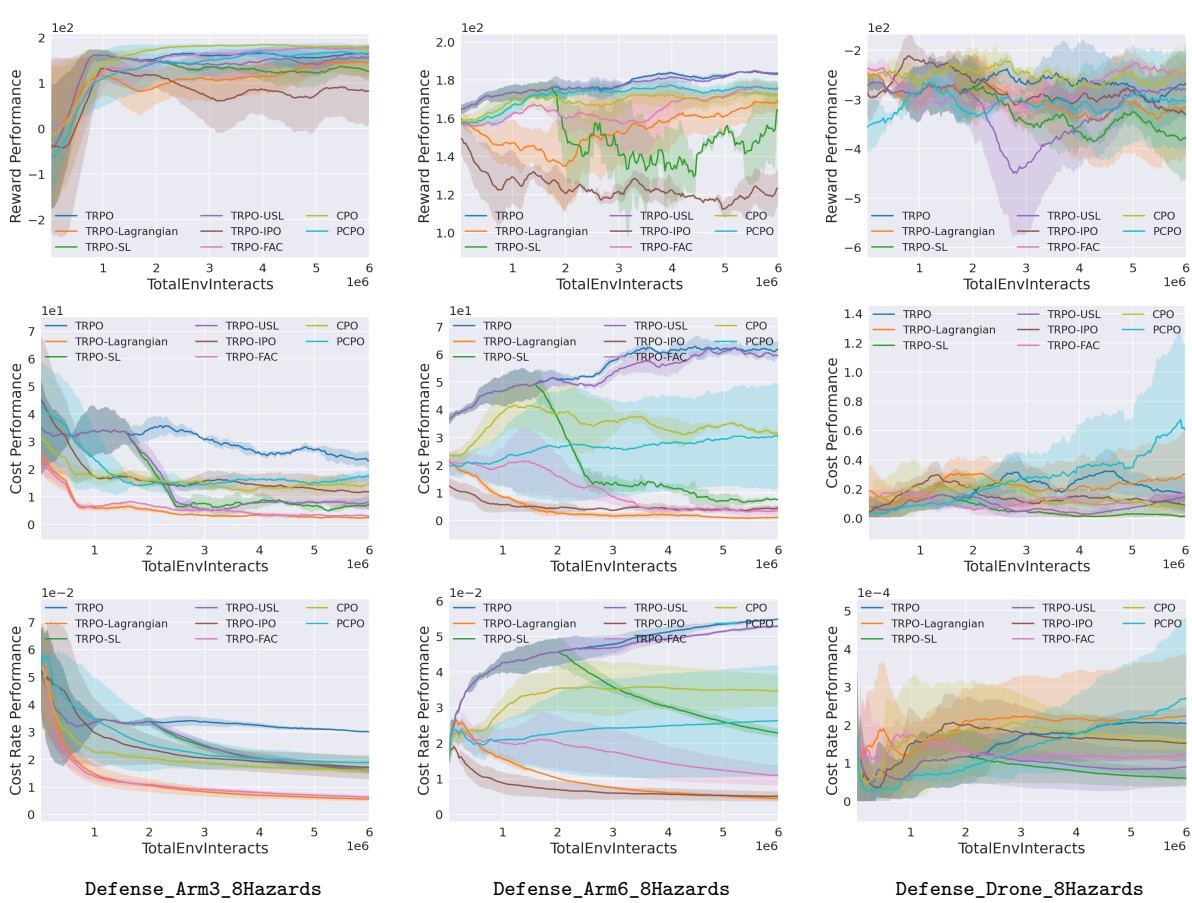

Defense_Arm3_8Hazards          Defense_Arm6_8Hazards          Defense_Drone_8Hazards

Figure 18: Defense_{Robot}_8Hazards

