# OpenReview forum: "GUARD: A Safe Reinforcement Learning Benchmark"
_TMLR — Accepted by TMLR_

### Review · Reviewer_84ZG · 2024-01-05

**Summary Of Contributions:**

The authors are primarily introducing a new benchmark for safe RL.
The article describes the more general set of tasks that are included a good range of the algorithms considered in this literature review.
Besides introducing this new benchmark, they perform an evaluation of the implemented algorithms on the provided tasks and shows the results as plots, and they have a limited discussion of take-aways from it.

**Audience:**

Yes

**Claims And Evidence:**

Yes

**Requested Changes:**

The main take-away in the discussion is formulated as  "Comparing
with them, hierarchical RL methods (i.e., TRPO-SL and TRPO-USL) can perform better at cost reduction,
but these methods excel at minimizing costs, they may sacrifice some degree of reward attainment in the
process." where them refers to constraint based methods. I think one should at least insert a "while" after "but".
Perhaps one could also expand upon analysis and discussion of results and see if one can offer more insights to the reader.

**Strengths And Weaknesses:**

Strength: The new evaluation suite seems to be a useful addition.
Weakness: Analysis and discussion of the results is limited.

---

> ### Author Response · Authors · 2024-01-22
> **Reply to Requested Changes**
>
> We extend our sincere gratitude for your thorough comments on our paper. We have updated the paper draft according to your comments, the new contents are highlighted in red color. Below, you will find our responses to your inquiries.
>
> **Weakness and Requested Change**: Expand upon analysis and discussion of results and see if one can offer more insights to the reader.
>
> **Answer**: Thank you for your valuable suggestions! We have diligently revamped the Experiment and Results section (Section 6) using a threefold strategy: (i) enhancing the clarity by providing more intricate details on the experimental setup, (ii) conducting a thorough evaluation of  safe Reinforcement Learning (RL) algorithms with respect to testing suites, addressing 8 key questions from diverse perspectives, and (iii) enriching the presentation with additional visualizations, including bar tables and spider web plots, to facilitate a comprehensive comparative analysis.

---

### Review · Reviewer_haVh · 2024-01-06

**Summary Of Contributions:**

This paper introduces GUARD, a software suite for evaluating safe RL algorithms. There are a number of benchmarks (from prior work) that have been included in the codebase.

**Audience:**

Yes

**Broader Impact Concerns:**

There are no broader impact concerns.

**Claims And Evidence:**

Yes

**Requested Changes:**

- Since the main contribution of the paper is the code repository, more so than the paper itself, it is very important that the authors address the points in Weaknesses above pertaining to the code base [critical for acceptance]
- There are a number of proofreading, formatting, and grammar issues that should be addressed, including a reference link overlaid on the number for page 3 and the top of page 4, "more latest" on page 10, etc.
- The link to the GitHub repository on page 26 is not anonymized, and even includes author names when visiting the page.
- Section C.2 on page 26 is missing.

**Strengths And Weaknesses:**

Strengths
- The paper addresses an important area, i.e., reliable and comprehensive baselines for safe reinforcement learning, and presents a software suite for being able to compare algorithms in a unified way.

Weaknesses
- The main contribution appears to be the code repository implementing the benchmark, more so than the paper itself. I believe that this is fine, provided that the code has: (1) very clear documentation about how to build on top of the existing implementations (as opposed to just re-running the benchmark experiments); (2) guidelines for contributing back to the repository; (3) an active plan for maintaining and expanding the code repository. Without these elements in place, the usefulness of the code for future research is relatively limited. The main question is whether this is a code release equivalent to what an original paper includes for reproducibility or is more robust and can be used for future experimentation and benchmarking in novel environments and settings, with different architectures, etc.
- The "Limitations and future work" section acknowledges a few important technical tasks including removing dependency on a library marked as deprecated "recently". However, this library was deprecated as of November 16, 2022 (https://github.com/openai/mujoco-py/commit/a13903b82f1ab316815e63b3575526005b3b2ae1). Therefore, it is important that there is a reasonable timeline for these changes to be made in order for GUARD to actually be useful as a modern benchmark suite.
- It is not clear how this codebase would work with other libraries; it appears to be highly customized to a particular set of environments, and is not sufficiently modular.
- The code has many inline comments but insufficient overall documentation. the code style overall is quite problematic and would be difficult for others to build upon. This includes pieces of code that have been commented out, TODO remarks in the code, extremely long classes with many `if` statements, etc.

---

> ### Author Response · Authors · 2024-01-22
> **Reply to Requested Changes**
>
> We express our sincere gratitude for dedicating your time and expertise to review our paper. Your insightful feedback has been invaluable to enhancing the quality of our work. In response to your suggestions, we have made significant improvements:
> - Code Repository Rework:
>    - Restructured and cleaned the code to enhance extensibility.
>    - Completely overhauled the README to provide clear guidance on usage and extension.
> - GUARD Expansions:
>     - Implemented several proposed expansions to the GUARD framework.
>     - Achieved promising results with the introduced modifications.
>
> The paper and supplementary materials have been thoroughly updated to reflect the latest changes. Below, you will find detailed responses to the questions and suggestions you provided.
>
> **Requested Change1 and Weakness1**: Code repository needs (i)  (1) very clear documentation about how to build on top of the existing implementations (as opposed to just re-running the benchmark experiments); (2) guidelines for contributing back to the repository; (3) an active plan for maintaining and expanding the code repository.
>
> **Answer1**: Thank you!
> - We have added *Build on top of GUARD* section in  README.md to comprehensively guide users on expanding the safe RL library and developing safe RL environments.
> -  We have added *Contributing to GUARD* section in README.md to provide explicit guidelines for contributors, outlining the process for contributing back to the repository.
> -  We have added *Maintaining and Expanding GUARD* section in README to list our detailed plans for maintaining the repository. Additionally, we proposed three expansion tasks and listed the proposed timeline. To support this expansion, we have started implementation and receive promising results, which are included in the supplementary materials under the video folder.
>
> We have printed the README as pdf for better readability, and it is located inside supplementary material.  $$ $$
>
>
>
> **Requested Change1 and Weakness2**: reasonable timeline to remove Mujoco-py
>
> **Answer2**: Thanks! During the last two weeks, we have diligently upgraded our simulation engine from Mujoco-py to Mujoco3 (replacing all associated APIs), which is the up-to-date Mujoco engine, making GUARD the modern benchmark suite. $$ $$
>
>
> **Requested Change1 and Weakness3**: How does the safe RL algorithm work with other environment libraries?
>
> **Answer3**: We have enhanced all algorithm implementations to ensure compatibility with various external libraries, provided they adhere to the Gym environment interface (https://gymnasium.farama.org/). Specifically, the ``create_env`` function for all algorithms has been updated to accommodate this notification. This information has been prominently featured in the README under the *Quick Start - Training* section for user awareness. $$ $$
>
>
> **Requested Change1 and Weakness4**: Clean code.
>
> **Answer4**: Thanks! We have removed all the unnecessary comments and TODO remarks. We also reorganized long functions into small modularized functions. $$ $$
>
>
> **Requested Change Other**: fix formatting, and grammar issues that should be addressed. Anonymize Github. Section C.2 on page 26 is missing.
>
> **Answer5**: we have fixed all the mentioned formatting issues, deleted the link, and referred to a Table in C.2.

---

> ### Comment · Reviewer_haVh · 2024-01-23
> **Re: Reply to Requested Changes**
>
> Thank you to the authors for your response and revisions. If possible, the authors should proofread the paper some more; a cursory look shows, for example, the title of Section C is misspelled (should be "Experiment Details").

---

> > ### Author Response · Authors · 2024-01-23
> > **Proofreading Completed**
> >
> > We appreciate your prompt response! We thoroughly reviewed the paper draft and meticulously proofread it multiple times. We addressed every grammar issue, typo, and potential confusing wording. The revised paper has been reuploaded. Thank you once again for your valuable time and effort!

---

### Review · Reviewer_3xHz · 2024-01-08

**Summary Of Contributions:**

This paper presents a new benchmark for GUARD safe reinforcement learning. GUARD includes a diverse set of robots with different level of control complexity. GUARD surveys and implements a range of SoTA algorithms. Details of the task design and evaluation of the algorithms are presented in the paper.

**Audience:**

Yes

**Claims And Evidence:**

No

**Requested Changes:**

I believe this paper can be greatly improved if the authors think about how to summarize and present their evaluation results better. For example, you can compute the average area under curve for each of the algorithm along different constraint types and construct a spider web plot to help the audience better understand the pros and cons of different camps of algorithms.

**Strengths And Weaknesses:**

Strengths:
- GUARD includes a range of different robotic platform and safe RL tasks
- GUARD implements a set of SoTA on-policy algorithms and ran evals on them to compare performances
- GUARD serves as a good survey for existing literature of safe RL

Weakness:
- GUARD does not propose conceptually different types of constraints that 1) do not existing in SafetyGym and 2) challenge existing safe RL algorithms (it seems most problems can be solved by at least some algorithm); -> for 2), the paper motivated by how safe RL is important for realworld applications, and maybe you could include one task that is more realistic (visual realistic and diverse)
- As a benchmark, it is important for GUARD to have comprehensive metrics evaluating different algorithms, probably along different axis of constraint complexity; while GUARD evaluated all the implementations of SoTA RL algorithms, it is hard to parse from the plots how different algorithms compare under GUARD

---

> ### Author Response · Authors · 2024-01-22
> **Reply to Weakness and Requested Changes**
>
> Thank you for your valuable feedback on our paper. We sincerely appreciate your time and effort in reviewing our work. We have updated the paper draft according to your comments, the new contents are highlighted in red color. Please find our answers to your questions below.
>
> **Weakness1**: include realistic robots.
>
> **Answer1**: Thanks! We have emphasized the integration of more realistic robots as a significant aspect of our future expansion plans, as outlined in the updated README. Concurrently, we have introduced two realistic robots into GUARD. Videos showcasing the trained tasks for these robots can be found in the `video/realistic_robot/` directory. $$ $$
>
>
> **Weakness 2 and Requested Change**: Summarize and present their evaluation results better, and construct a spider web plot to help the audience better understand the pros and cons of different camps of algorithms.
>
> **Answer2**: Thank you for your suggestion! We have completely rewrote the Experiment and Results section (Section 6) through a threefold approach: (i) offering more details on the experimental setup, (ii) conducting a comprehensive evaluation of the pros and cons of safe Reinforcement Learning (RL) algorithms by addressing 8 key questions from various perspectives, and (iii) incorporating additional visualizations such as bar tables and spider web plots for a more in-depth comparative analysis.

---

### Decision · Action_Editor_BLrW · 2024-02-21

**Recommendation:** Accept as is

**Comment:**

AI Safety is a topic of surging interest, and RL safety in particular is of significant importance in areas like Robotics. As such, widely adopted benchmarks will help steer the field forward. This paper was not recognized to be particularly novel, but the reviewers agree that the released code on GitHub and documentation of evals would be valuable for the field.

**Audience:**

Yes. While the paper does not necessarily go beyond other benchmarks like SafeGym in terms of novelty, the codebase would be a valuable resource for the RL safety community to build upon.

**Claims And Evidence:**

The primary contribution of this paper is a new benchmark for safe RL, with increasing levels of complexity on multiple robot platforms. The paper comes with a comprehensive evaluation of algorithms clarifying what the state of the art is.